# End-to-end Learning of LDA by Mirror-Descent Back Propagation over a Deep Architecture

**Jianshu Chen**∗, **Ji He**†, **Yelong Shen**∗, **Lin Xiao**∗, **Xiaodong He**∗, **Jianfeng Gao**∗, **Xinying Song**∗ **and Li Deng**∗

∗Microsoft Research, Redmond, WA 98052, USA,
{jianshuc,yeshen,lin.xiao,xiaohe,jfgao,xinson,deng}@microsoft.com
†Department of Electrical Engineering, University of Washington, Seattle, WA 98195, USA,
jvking@uw.edu

## Abstract

We develop a fully discriminative learning approach for supervised Latent Dirichlet Allocation (LDA) model using Back Propagation (i.e., BP-sLDA), which maximizes the posterior probability of the prediction variable given the input document. Different from traditional variational learning or Gibbs sampling approaches, the proposed learning method applies (i) the mirror descent algorithm for maximum a posterior inference and (ii) back propagation over a deep architecture together with stochastic gradient/mirror descent for model parameter estimation, leading to scalable and end-to-end discriminative learning of the model. As a byproduct, we also apply this technique to develop a new learning method for the traditional unsupervised LDA model (i.e., BP-LDA). Experimental results on three real-world regression and classification tasks show that the proposed methods significantly outperform the previous supervised topic models, neural networks, and is on par with deep neural networks.

## 1 Introduction

Latent Dirichlet Allocation (LDA) [5], among various forms of topic models, is an important probabilistic generative model for analyzing large collections of text corpora. In LDA, each document is modeled as a collection of words, where each word is assumed to be generated from a certain topic drawn from a topic distribution. The topic distribution can be viewed as a latent representation of the document, which can be used as a feature for prediction purpose (e.g., sentiment analysis). In particular, the inferred topic distribution is fed into a separate classifier or regression model (e.g., logistic regression or linear regression) to perform prediction. Such a separate learning structure usually significantly restricts the performance of the algorithm. For this purpose, various supervised topic models have been proposed to model the documents jointly with the label information. In [4], variational methods was applied to learn a supervised LDA (sLDA) model by maximizing the lower bound of the joint probability of the input data and the labels. The DiscLDA method developed in [15] learns the transformation matrix from the latent topic representation to the output in a discriminative manner, while learning the topic to word distribution in a generative manner similar to the standard LDA. In [26], max margin supervised topic models are developed for classification and regression, which are trained by optimizing the sum of the variational bound for the log marginal likelihood and an additional term that characterizes the prediction margin. These methods successfully incorporate the information from both the input data and the labels, and showed better performance in prediction compared to the vanilla LDA model.

One challenge in LDA is that the exact inference is intractable, i.e., the posterior distribution of the topics given the input document cannot be evaluated explicitly. For this reason, various approximate

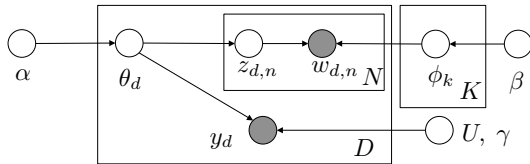

Figure 1: Graphical representation of the supervised LDA model. Shaded nodes are observables.

inference methods are proposed, such as variational learning [4, 5, 26] and Gibbs sampling [9, 27], for computing the approximate posterior distribution of the topics. In this paper, we will show that, although the full posterior probability of the topic distribution is difficult, its maximum a posteriori (MAP) inference, as a simplified problem, is a convex optimization problem when the Dirichlet parameter satisfies certain conditions, which can be solved efficiently by the mirror descent algorithm (MDA) [2, 18, 21]. Indeed, Sontag and Roy [19] pointed out that the MAP inference problem of LDA in this situation is polynomial-time and can be solved by an exponentiated gradient method, which shares a same form as our mirror-descent algorithm with constant step-size. Nevertheless, different from [19], which studied the inference problem alone, our focus in this paper is to integrate back propagation with mirror-descent algorithm to perform fully discriminative training of supervised topic models, as we proceed to explain below.

Among the aforementioned methods, one training objective of the supervised LDA model is to maximize the joint likelihood of the input and the output variables [4]. Another variant is to maximize the sum of the log likelihood (or its variable bound) and a prediction margin [26, 27]. Moreover, the DiscLDA optimizes part of the model parameters by maximizing the marginal likelihood of the input variables, and optimizes the other part of the model parameters by maximizing the conditional likelihood. For this reason, DiscLDA is not a fully discriminative training of all the model parameters. In this paper, we propose a fully discriminative training of all the model parameters by maximizing the posterior probability of the output given the input document. We will show that the discriminative training can be performed in a principled manner by naturally integrating the back-propagation with the MDA-based exact MAP inference. To our best knowledge, this paper is the first work to perform a fully end-to-end discriminative training of supervised topic models. Discriminative training of generative model is widely used and usually outperforms standard generative training in prediction tasks [3, 7, 12, 14, 25]. As pointed out in [3], discriminative training increases the robustness against the mismatch between the generative model and the real data. Experimental results on three real-world tasks also show the superior performance of discriminative training.

In addition to the aforementioned related studies on topic models [4, 15, 26, 27], there have been another stream of work that applied empirical risk minimization to graphical models such as Markov Random Field and nonnegative matrix factorization [10, 20]. Specifically, in [20], an approximate inference algorithm, belief propagation, is used to compute the belief of the output variables, which is further fed into a decoder to produce the prediction. The approximate inference and the decoder are treated as an entire black-box decision rule, which is tuned jointly via back propagation. Our work is different from the above studies in that we use an MAP inference based on optimization theory to motivate the discriminative training from a principled probabilistic framework.

## 2  Smoothed Supervised LDA Model

We consider the smoothed supervised LDA model in Figure 1. Let $K$ be the number of topics, $N$ be the number of words in each document, $V$ be the vocabulary size, and $D$ be the number of documents in the corpus. The generative process of the model in Figure 1 can be described as:

1. For each document $d$, choose the topic proportions according to a Dirichlet distribution: $\theta_d \sim p(\theta_d|\alpha) = \mathrm{Dir}(\alpha)$, where $\alpha$ is a $K \times 1$ vector consisting of nonnegative components.

2. Draw each column $\phi_k$ of a $V \times K$ matrix $\Phi$ independently from an exchangeable Dirichlet distribution: $\phi_k \sim \mathrm{Dir}(\beta)$ (i.e., $\Phi \sim p(\Phi|\beta)$), where $\beta > 0$ is the smoothing parameter.

3. To generate each word $w_{d,n}$:

(a) Choose a topic $z_{d,n} \sim p(z_{d,n}|\theta_d) = \text{Multinomial}(\theta_d)$. [1]

(b) Choose a word $w_{d,n} \sim p(w_{d,n}|z_{d,n}, \Phi) = \text{Multinomial}(\phi_{z_{d,n}})$.

4. Choose the $C \times 1$ response vector: $y_d \sim p(y_d|\theta, U, \gamma)$.

(a) In regression, $p(y_d|\theta_d, U, \gamma) = N(U\theta_d, \gamma^{-1})$, where $U$ is a $C \times K$ matrix consisting of regression coefficients.

(b) In multi-class classification, $p(y_d|\theta_d, U, \gamma) = \text{Multinomial}\big(\text{Softmax}(\gamma U\theta_d)\big)$, where the softmax function is defined as $\text{Softmax}(x)_c = \frac{e^{x_c}}{\sum_{c'=1}^{C} e^{x_{c'}}}, c = 1, \dots, C$.

Therefore, the entire model can be described by the following joint probability

$$p(\Phi|\beta) \prod_{d=1}^{D} \Big[ \underbrace{p(y_d|\theta_d, U, \gamma) \cdot p(\theta_d|\alpha) \cdot p(w_{d,1:N}|z_{d,1:N}, \Phi) \cdot p(z_{d,1:N}|\theta_d)}_{\triangleq p(y_d, \theta_d, w_{d,1:N}, z_{d,1:N}|\Phi, U, \alpha, \gamma)} \Big] \qquad (1)$$

where $w_{d,1:N}$ and $z_{d,1:N}$ denotes all the words and the associated topics, respectively, in the $d$-th document. Note that the model in Figure 1 is slightly different from the one proposed in [4], where the response variable $y_d$ in Figure 1 is coupled with $\theta_d$ instead of $z_{d,1:N}$ as in [4]. Blei and Mcauliffe also pointed out this choice as an alternative in [4]. This modification will lead to a differentiable end-to-end cost trainable by back propagation with superior prediction performance.

To develop a fully discriminative training method for the model parameters $\Phi$ and $U$, we follow the argument in [3], which states that the discriminative training is also equivalent to maximizing the joint likelihood of a new model family with an additional set of parameters:

$$\arg\max_{\Phi, U, \tilde{\Phi}} p(\Phi|\beta)p(\tilde{\Phi}|\beta) \prod_{d=1}^{D} p(y_d|w_{d,1:N}, \Phi, U, \alpha, \gamma) \prod_{d=1}^{D} p(w_{d,1:N}|\tilde{\Phi}, \alpha) \qquad (2)$$

where $p(w_{d,1:N}|\tilde{\Phi}, \alpha)$ is obtained by marginalizing $p(y_d, \theta_d, w_{d,1:N}, z_{d,1:N}|\Phi, U, \alpha, \gamma)$ in (1) and replace $\Phi$ with $\tilde{\Phi}$. The above problem (2) decouples into

$$\arg\max_{\Phi, U} \Big[ \ln p(\Phi|\beta) + \sum_{d=1}^{D} \ln p(y_d|w_{d,1:N}, \Phi, U, \alpha, \gamma) \Big] \qquad (3)$$

$$\arg\max_{\tilde{\Phi}} \Big[ \ln p(\tilde{\Phi}|\beta) + \sum_{d=1}^{D} \ln p(w_{d,1:N}|\tilde{\Phi}, \alpha) \Big] \qquad (4)$$

which are the discriminative learning problem of supervised LDA (Eq. (3)), and the unsupervised learning problem of LDA (Eq. (4)), respectively. We will show that both problems can be solved in a unified manner using a new MAP inference and back propagation.

## 3 Maximum A Posterior (MAP) Inference

We first consider the inference problem in the smoothed LDA model. For the supervised case, the main objective is to infer $y_d$ given the words $w_{d,1:N}$ in each document $d$, i.e., computing

$$p(y_d|w_{d,1:N}, \Phi, U, \alpha, \gamma) = \int_{\theta_d} p(y_d|\theta_d, U, \gamma) p(\theta_d|w_{d,1:N}, \Phi, \alpha) d\theta_d \qquad (5)$$

where the probability $p(y_d|\theta_d, U, \gamma)$ is known (e.g., multinomial or Gaussian for classification and regression problems — see Section 2). The main challenge is to evaluate $p(\theta_d|w_{d,1:N}, \Phi, \alpha)$, i.e., infer the topic proportion given each document, which is also the important inference problem in the unsupervised LDA model. However, it is well known that the exact evaluation of the posterior probability $p(\theta_d|w_{d,1:N}, \Phi, \alpha)$ is intractable [4, 5, 9, 15, 26, 27]. For this reason, various approximate inference methods, such as variational inference [4, 5, 15, 26] and Gibbs sampling [9, 27],

have been proposed to compute the approximate posterior probability. In this paper, we take an alternative approach for inference; given each document $d$, we only seek a point (MAP) estimate of $\theta_d$, instead of its full (approximate) posterior probability. The major motivation is that, although the full posterior probability of $\theta_d$ is difficult, its MAP estimate, as a simplified problem, is more tractable (and it is a convex problem under certain conditions). Furthermore, with the MAP estimate of $\theta_d$, we can infer the prediction variable $y_d$ according to the following approximation from (5):

$$p(y_d|w_{d,1:N}, \Phi, U, \alpha, \gamma) = \mathbb{E}_{\theta_d|w_{d,1:N}} \left[ p(y_d|\theta_d, U, \gamma) \right] \approx p(y_d|\hat{\theta}_{d|w_{d,1:N}}, U, \gamma) \qquad (6)$$

where $\mathbb{E}_{\theta_d|w_{d,1:N}}$ denotes the conditional expectation with respect to $\theta_d$ given $w_{d,1:N}$, and the expectation is sampled by the MAP estimate, $\hat{\theta}_{d|w_{d,1:N}}$, of $\theta_d$ given $w_{d,1:N}$, defined as

$$\hat{\theta}_{d|w_{d,1:N}} = \arg \max_{\theta_d} p(\theta_d|w_{d,1:N}, \Phi, \alpha, \beta) \qquad (7)$$

The approximation gets more precise when $p(\theta_d|w_{d,1:N}, \Phi, \alpha, \beta)$ becomes more concentrated around $\hat{\theta}_{d|w_{d,1:N}}$. Experimental results on several real datasets (Section 5) show that the approximation (6) provides excellent prediction performance.

Using the Bayesian rule $p(\theta_d|w_{d,1:N}, \Phi, \alpha) = p(\theta_d|\alpha)p(w_{d,1:N}|\theta_d, \Phi)/p(w_{d,1:N}|\Phi, \alpha)$ and the fact that $p(w_{d,1:N}|\Phi, \alpha)$ is independent of $\theta_d$, we obtain the equivalent form of (7) as

$$\hat{\theta}_{d|w_{d,1:N}} = \arg \max_{\theta_d \in \mathcal{P}_K} \left[ \ln p(\theta_d|\alpha) + \ln p(w_{d,1:N}|\theta_d, \Phi) \right] \qquad (8)$$

where $\mathcal{P}_K = \{\theta \in \mathbb{R}^K : \theta_j \geq 0, \sum_{j=1}^K \theta_j = 1\}$ denotes the $(K-1)$-dimensional probability simplex, $p(\theta_d|\alpha)$ is the Dirichlet distribution, and $p(w_{d,1:N}|\theta_d, \Phi)$ can be computed by integrating $p(w_{d,1:N}, z_{d,1:N}|\theta_d, \Phi) = \prod_{n=1}^N p(w_{d,n}|z_{d,n}, \Phi)p(z_{d,n}|\theta_d)$ over $z_{d,1:N}$, which leads to (derived in Section A of the supplementary material)

$$p(w_{d,1:N}|\theta_d, \Phi) = \prod_{v=1}^V \left( \sum_{j=1}^K \theta_{d,j} \Phi_{vj} \right)^{x_{d,v}} = p(x_d|\theta_d, \Phi) \qquad (9)$$

where $x_{d,v}$ denotes the term frequency of the $v$-th word (in vocabulary) inside the $d$-th document, and $x_d$ denotes the $V$-dimensional bag-of-words (BoW) vector of the $d$-th document. Note that $p(w_{d,1:N}|\theta_d, \Phi)$ depends on $w_{d,1:N}$ only via the BoW vector $x_d$, which is the sufficient statistics. Therefore, we use $p(x_d|\theta_d, \Phi)$ and $p(w_{d,1:N}|\theta_d, \Phi)$ interchangeably from now on. Substituting the expression of Dirichlet distribution and (9) into (8), we get

$$\begin{aligned} \hat{\theta}_{d|w_{d,1:N}} &= \arg \max_{\theta_d \in \mathcal{P}_K} \left[ x_d^T \ln(\Phi\theta_d) + (\alpha - \mathbb{1})^T \ln \theta_d \right] \\ &= \arg \min_{\theta_d \in \mathcal{P}_K} \left[ -x_d^T \ln(\Phi\theta_d) - (\alpha - \mathbb{1})^T \ln \theta_d \right] \end{aligned} \qquad (10)$$

where we dropped the terms independent of $\theta_d$, and $\mathbb{1}$ denotes an all-one vector. Note that when $\alpha \geq 1$ ($\alpha > 1$), the optimization problem (10) is (strictly) convex and is non-convex otherwise.

## 3.1 Mirror Descent Algorithm for MAP Inference

An efficient approach to solving the constrained optimization problem (10) is the mirror descent algorithm (MDA) with Bregman divergence chosen to be generalized Kullback-Leibler divergence [2, 18, 21]. Specifically, let $f(\theta_d)$ denote the cost function in (10), then the MDA updates the MAP estimate of $\theta_d$ iteratively according to:

$$\theta_{d,\ell} = \arg \min_{\theta_d \in \mathcal{P}_K} \left[ f(\theta_{d,\ell-1}) + [\nabla_{\theta_d} f(\theta_{d,\ell-1})]^T (\theta_d - \theta_{d,\ell-1}) + \frac{1}{T_{d,\ell}} \Psi(\theta_d, \theta_{d,\ell-1}) \right] \qquad (11)$$

$\theta_{d,\ell}$ denotes the estimate of $\theta_{d,\ell}$ at the $\ell$-th iteration, $T_{d,\ell}$ denotes the step-size of MDA, and $\Psi(x, y)$ is the Bregman divergence chosen to be $\Psi(x, y) = x^T \ln(x/y) - \mathbb{1}^T x + \mathbb{1}^T y$. The argmin in (11) can be solved in closed-form (see Section B of the supplementary material) as

$$\theta_{d,\ell} = \frac{1}{C_\theta} \cdot \theta_{d,\ell-1} \odot \exp \left( T_{d,\ell} \left[ \Phi^T \frac{x_d}{\Phi\theta_{d,\ell-1}} + \frac{\alpha - \mathbb{1}}{\theta_{d,\ell-1}} \right] \right), \quad \ell = 1, \ldots, L, \quad \theta_{d,0} = \frac{1}{K}\mathbb{1} \qquad (12)$$

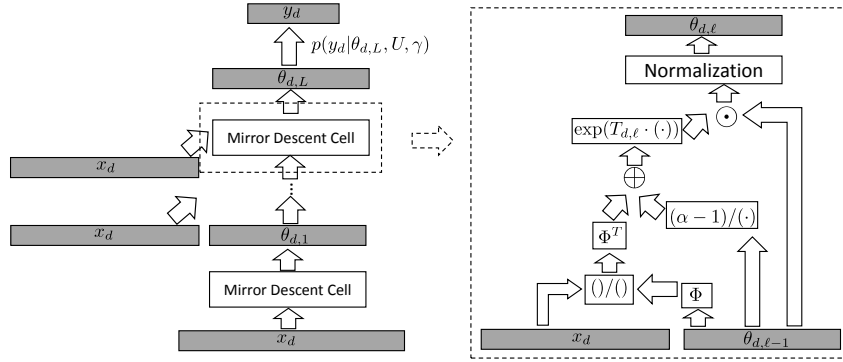

Figure 2: Layered deep architecture for computing $p(y_d|w_{d,1:N}, \Phi, U, \alpha, \gamma)$, where $()/()$ denotes element-wise division, $\odot$ denotes Hadamard product, and $\exp()$ denotes element-wise exponential.

where $C_\theta$ is a normalization factor such that $\theta_{d,\ell}$ adds up to one, $\odot$ denotes Hadamard product, $L$ is the number of MDA iterations, and the divisions in (12) are element-wise operations. Note that the recursion (12) naturally enforces each $\theta_{d,\ell}$ to be on the probability simplex. The MDA step-size $T_{d,\ell}$ can be either constant, i.e., $T_{d,\ell} = T$, or adaptive over iterations and samples, determined by line search (see Section C of the supplementary material). The computation complexity in (12) is low since most computations are sparse matrix operations. For example, although by itself $\Phi\theta_{d,\ell-1}$ in (12) is a dense matrix multiplication, we only need to evaluate the elements of $\Phi\theta_{d,\ell-1}$ at the positions where the corresponding elements of $x_d$ are nonzero, because all other elements of $x_d/\Phi\theta_{d,\ell-1}$ is known to be zero. Overall, the computation complexity in each iteration of (12) is $O(\text{nTok} \cdot K)$, where $\text{nTok}$ denotes the number of unique tokens in the document. In practice, we only use a small number of iterations, $L$, in (12) and use $\theta_{d,L}$ to approximate $\hat{\theta}_{d|w_{d,1:N}}$ so that (6) becomes

$$p(y_d|w_{d,1:N}, \Phi, U, \alpha, \gamma) \approx p(y_d|\theta_{d,L}, U, \gamma) \qquad (13)$$

In summary, the inference of $\theta_d$ and $y_d$ can be implemented by the layered architecture in Figure 2, where the top layer infers $y_d$ using (13) and the MDA layers infer $\theta_d$ iteratively using (12). Figure 2 also implies that the the MDA layers act as a feature extractor by generating the MAP estimate $\theta_{d,L}$ for the output layer. Our end-to-end learning strategy developed in the next section jointly learns the model parameter $U$ at the output layer and the model parameter $\Phi$ at the feature extractor layers to maximize the posterior of the prediction variable given the input document.

## 4   Learning by Mirror-Descent Back Propagation

We now consider the supervised learning problem (3) and the unsupervised learning problem (4), respectively, using the developed MDA-based MAP inference. We first consider the supervised learning problem. With (13), the discriminative learning problem (3) can be approximated by

$$\arg\min_{\Phi,U} \left[ -\ln p(\Phi|\beta) - \sum_{d=1}^{D} \ln p(y_d|\theta_{d,L}, U, \gamma) \right] \qquad (14)$$

which can be solved by stochastic mirror descent (SMD). Note that the cost function in (14) depends on $U$ explicitly through $p(y_d|\theta_{d,L}, U, \gamma)$, which can be computed directly from its definition in Section 2. On the other hand, the cost function in (14) depends on $\Phi$ implicitly through $\theta_{d,L}$. From Figure 2, we observe that $\theta_{d,L}$ not only depends on $\Phi$ explicitly (as indicated in the MDA block on the right-hand side of Figure 2) but also depends on $\Phi$ implicitly via $\theta_{d,L-1}$, which in turn depends on $\Phi$ both explicitly and implicitly (through $\theta_{d,L-2}$) and so on. That is, the dependency of the cost function on $\Phi$ is in a layered manner. Therefore, we devise a back propagation procedure to efficiently compute its gradient with respect to $\Phi$ according to the mirror-descent graph in Figure 2, which back propagate the error signal through the MDA blocks at different layers. The gradient formula and the implementation details of the learning algorithm can be found in Sections C–D in the supplementary material.

For the unsupervised learning problem (4), the gradient of $\ln p(\tilde{\Phi}|\beta)$ with respect to $\tilde{\Phi}$ assumes the same form as that of $\ln p(\Phi|\beta)$. Moreover, it can be shown that the gradient of $\ln p(w_{d,1:N}|\tilde{\Phi}, \alpha, \gamma)$

with respect $\tilde{\Phi}$ can be expressed as (see Section E of the supplementary material):

$$\frac{\partial \ln p(w_{d,1:N}|\tilde{\Phi},\alpha)}{\partial \tilde{\Phi}} = \mathbb{E}_{\theta_d|x_d}\left\{\frac{\partial}{\partial \tilde{\Phi}}\ln p(x_d|\theta_d,\tilde{\Phi})\right\} \overset{(a)}{\approx} \frac{\partial}{\partial \tilde{\Phi}}\ln p(x_d|\theta_{d,L},\tilde{\Phi}) \qquad (15)$$

where $p(x_d|\theta_d,\tilde{\Phi})$ assumes the same form as (9) except $\Phi$ is replaced by $\tilde{\Phi}$. The expectation is evaluated with respect to the posterior probability $p(\theta_d|w_{d,1:N},\tilde{\Phi},\alpha)$, and is sampled by the MAP estimate of $\theta_d$ in step (a). $\theta_{d,L}$ is an approximation of $\hat{\theta}_{d|w_{d,1:N}}$ computed via (12) and Figure 2.

## 5 Experiments

### 5.1 Description of Datasets and Baselines

We evaluated our proposed supervised learning (denoted as BP-sLDA) and unsupervised learning (denoted as BP-LDA) methods on three real-world datasets. The first dataset we use is a large-scale dataset built on Amazon movie reviews (AMR) [16]. The data set consists of 7.9 million movie reviews (1.48 billion words) from Amazon, written by 889,176 users, on a total of 253,059 movies. For text preprocessing we removed punctuations and lowercasing capital letters. A vocabulary of size 5,000 is built by selecting the most frequent words. (In another setup, we keep the full vocabulary of 701K.) Same as [24], we shifted the review scores so that they have zero mean. The task is formulated as a regression problem, where we seek to predict the rating score using the text of the review. Second, we consider a multi-domain sentiment (MultiSent) classification task [6], which contains a total 342,104 reviews on 25 types of products, such as apparel, electronics, kitchen and housewares. The task is formulated as a binary classification problem to predict the polarity (positive or negative) of each review. Likewise, we preprocessed the text by removing punctuations and lowercasing capital letters, and built a vocabulary of size 1,000 from the most frequent words. In addition, we also conducted a second binary text classification experiment on a large-scale proprietary dataset for business-centric applications (1.2M documents and vocabulary size of 128K).

The baseline algorithms we considered include Gibbs sampling (Gibbs-LDA) [17], logistic/linear regression on bag-of-words, supervised-LDA (sLDA) [4], and MedLDA [26], which are implemented either in C++ or Java. And our proposed algorithms are implemented in C#.[2] For BP-LDA and Gibbs-LDA, we first train the models in an unsupervised manner, and then generate per-document topic proportion $\theta_d$ as their features in the inference steps, on top of which we train a linear (logistic) regression model on the regression (classification) tasks.

### 5.2 Prediction Performance

We first evaluate the prediction performance of our models and compare them with the traditional (supervised) topic models. Since the training of the baseline topic models takes much longer time than BP-sLDA and BP-LDA (see Figure 5), we compare their performance on two smaller datasets, namely a subset (79K documents) of AMR (randomly sampled from the 7.9 million reviews) and the MultiSent dataset (342K documents), which are all evaluated with 5-fold cross validation. For AMR regression, we use the predictive $R^2$ to measure the prediction performance, defined as: $pR^2 = 1 - (\sum_d (y_d^o - y_d)^2)/(\sum_d (y_d^o - \bar{y}_d^o)^2)$, where $y_d^o$ denotes the label of the $d$-th document in the heldout (out-of-fold) set during the 5-fold cross validation, $\bar{y}_d^o$ is the mean of all $y_d^o$ in the heldout set, and $y_d$ is the predicted value. The $pR^2$ scores of different models with varying number of topics are shown in Figure 3(a). Note that the BP-sLDA model outperforms the other baselines with large margin. Moreover, the unsupervised BP-LDA model outperforms the unsupervised LDA model trained by Gibbs sampling (Gibbs-LDA). Second, on the MultiSent binary classification task, we use the area-under-the-curve (AUC) of the operating curve of probability of correct positive versus probability of false positive as our performance metric, which are shown in Figure 3(b). It also shows that BP-sLDA outperforms other methods and that BP-LDA outperforms the Gibbs-LDA model.

Next, we compare our BP-sLDA model with other strong discriminative models (such as neural networks) by conducting two large-scale experiments: (i) regression task on AMR full dataset (7.9M documents) and (ii) binary classification task on the proprietary business-centric dataset (1.2M documents). For the large-scale AMR regression, we can see that $pR^2$ improves significantly compared

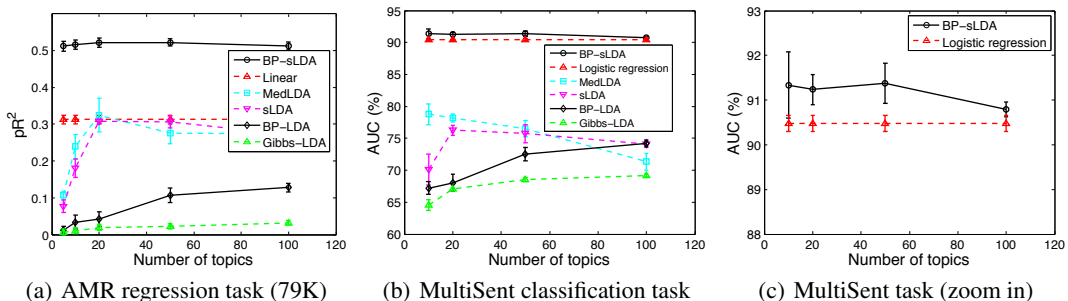

(a) AMR regression task (79K)  (b) MultiSent classification task  (c) MultiSent task (zoom in)

Figure 3: Prediction performance on AMR regression task (measured in $pR^2$) and MultiSent classification task (measured in AUC). Higher score is better for both, with perfect value being one.

Table 1: $pR^2$ (in percentage) on full AMR data (7.9M documents). The standard deviations in the parentheses are obtained from 5-fold cross validation.

| Number of topics | 5 | 10 | 20 | 50 | 100 | 200 |
|---|---|---|---|---|---|---|
| Linear Regression (voc5K) | 38.4 (0.1) | | | | | |
| Neural Network (voc5K) | 59.0 (0.1) | 61.0 (0.1) | 62.3 (0.4) | 63.5 (0.7) | 63.1 (0.8) | 63.5 (0.4) |
| BP-sLDA ($\alpha = 1.001$, voc5K) | 61.4 (0.1) | 65.3 (0.3) | 69.1 (0.2) | 74.7 (0.3) | 74.3 (2.4) | **78.3** (1.1) |
| BP-sLDA ($\alpha = 0.5$, voc5K) | 54.7 (0.1) | 54.5 (1.2) | 57.0 (0.2) | 61.3 (0.3) | 67.1 (0.1) | 74.5 (0.2) |
| BP-sLDA ($\alpha = 0.1$, voc5K) | 53.3 (2.8) | 56.1 (0.1) | 58.4 (0.1) | 64.1 (0.1) | 70.6 (0.3) | 75.7 (0.2) |
| Linear Regression (voc701K) | 41.5 (0.2) | | | | | |
| BP-sLDA ($\alpha=1.001$,voc701K) | 69.8 (0.2) | 74.3 (0.3) | 78.5 (0.2) | 83.6 (0.6) | 80.1 (0.9) | **84.7** (2.8) |

to the best results on the 79K dataset shown in Figure 3(a), and also significantly outperform the neural network models with same number of model parameters. Moreover, the best deep neural network ($200 \times 200$ in hidden layers) gives $pR^2$ of $76.2\%(\pm 0.6\%)$, which is worse than $78.3\%$ of BP-sLDA. In addition, BP-sLDA also significantly outperforms Gibbs-sLDA [27], Spectral-sLDA [24], and the Hybrid method (Gibbs-sLDA initialized with Spectral-sLDA) [24], whose $pR^2$ scores (reported in [24]) are between $10\%$ and $20\%$ for $5 \sim 10$ topics (and deteriorate when further increasing the topic number). The results therein are obtained under same setting as this paper. To further demonstrate the superior performance of BP-sLDA on the large vocabulary scenario, we trained BP-sLDA on full vocabulary (701K) AMR and show the results in Table 1, which are even better than the 5K vocabulary case. Finally, for the binary text classification task on the proprietary dataset, the AUCs are given in Table 2, where BP-sLDA (200 topics) achieves $31\%$ and $18\%$ relative improvements over logistic regression and neural network, respectively. Moreover, on this task, BP-sLDA is also on par with the best DNN (a larger model consisting of $200 \times 200$ hidden units with dropout), which achieves an AUC of 93.60.

## 5.3 Analysis and Discussion

We now analyze the influence of different hyper parameters on the prediction performance. Note from Figure 3(a) that, when we increase the number of topics, the $pR^2$ score of BP-sLDA first improves and then slightly deteriorates after it goes beyond 20 topics. This is most likely to be caused by overfitting on the small dataset (79K documents), because the BP-sLDA models trained on the full 7.9M dataset produce much higher $pR^2$ scores (Table 1) than that on the 79K dataset and keep improving as the model size (number of topics) increases. To understand the influence of the mirror descent steps on the prediction performance, we plot in Figure 4(a) the $pR^2$ scores of BP-sLDA on the 7.9M AMR dataset for different values of mirror-descent steps $L$. When $L$ increases, for small models ($K = 5$ and $K = 20$), the $pR^2$ score remains the same, and, for a larger model ($K = 100$), the $pR^2$ score first improves and then remain the same. One explanation for this phenomena is that larger $K$ implies that the inference problem (10) becomes an optimization problem of higher dimension, which requires more mirror descent iterations. Moreover, the mirror-descent back propagation, as an end-to-end training of the prediction output, would compensate the imperfection caused by the limited number of inference steps, which makes the performance insensitive to $L$ once it is large enough. In Figure 4(b), we plot the percentage of the dominant

Table 2: AUC (in percentage) on the business-centric proprietary data (1.2M documents, 128K vocabulary). The standard deviations in the parentheses are obtained from five random initializations.

| Number of topics | 5 | 10 | 20 | 50 | 100 | 200 |
|---|---|---|---|---|---|---|
| Logistic Regression | 90.56 (0.00) | | | | | |
| Neural Network | 90.95 (0.07) | 91.25 (0.05) | 91.32 (0.23) | 91.54 (0.11) | 91.90 (0.05) | 91.98 (0.05) |
| BP-sLDA | 92.02 (0.02) | 92.21 (0.03) | 92.35 (0.07) | 92.58 (0.03) | 92.82 (0.07) | **93.50** (0.06) |

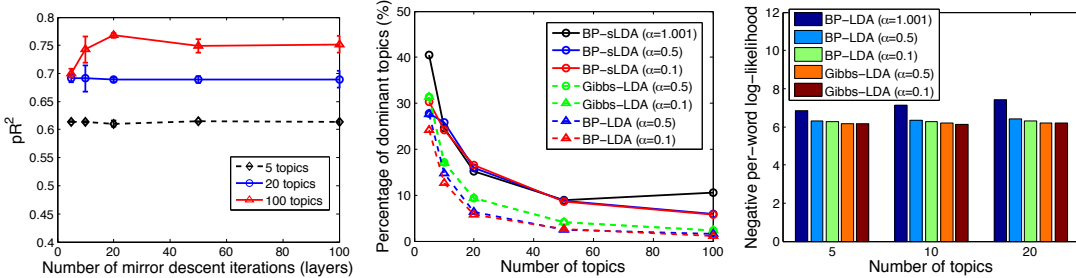

(a) Influence of MDA iterations $L$    (b) Sparsity of the topic distribution    (c) Per-word log-likelihoods

Figure 4: Analysis of the behaviors of BP-sLDA and BP-LDA models.

topics (which add up to 90% probability) on AMR, which shows that BP-sLDA learns sparse topic distribution even when $\alpha = 1.001$ and obtains sparser topic distribution with smaller $\alpha$ (i.e., 0.5 and 0.1). In Figure 4(c), we evaluate the per-word log-likelihoods of the unsupervised models on AMR dataset using the method in [23]. The per-word log-likelihood of BP-LDA with $\alpha = 1.001$ is worse than the case of $\alpha = 0.5$ and $\alpha = 0.1$ for Gibbs-LDA, although its prediction performance is better. This suggests the importance of the Dirichlet prior in text modeling [1, 22] and a potential tradeoff between the text modeling performance and the prediction performance.

## 5.4 Efficiency in Computation Time

To compare the efficiency of the algorithms, we show the training time of different models on the AMR dataset (79K and 7.9M) in Figure 5, which shows that our algorithm scales well with respect to increasing model size (number of topics) and increasing number of data samples.

## 6 Conclusion

We have developed novel learning approaches for supervised LDA models, using MAP inference and mirror-descent back propagation, which leads to an end-to-end discriminative training. We evaluate the prediction performance of the model on three real-world regression and classification tasks. The results show that the discriminative training significantly improves the performance of the supervised LDA model relative to previous learning methods.

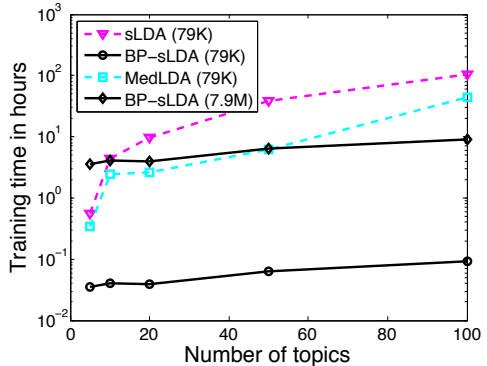

Figure 5: Training time on the AMR dataset. (Tested on Intel Xeon E5-2680 2.80GHz.)

Future works include (i) exploring faster algorithms for the MAP inference (e.g., accelerated mirror descent), (ii) developing semi-supervised learning of LDA using the framework from [3], and (iii) learning $\alpha$ from data. Finally, also note that the layered architecture in Figure 2 could be viewed as a deep feedforward neural network [11] with structures designed from the topic model in Figure 1. This opens up a new direction of combining the strength of both generative models and neural networks to develop new deep learning models that are scalable, interpretable and having high prediction performance for text understanding and information retrieval [13].

## Footnotes

[1]We will represent all the multinomial variables by a one-hot vector that has a single component equal to one at the position determined by the multinomial variable and all other components being zero.

[2]A third-party code is available online at `https://github.com/jvking/bp-lda`.

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
