[Supplementary Material]

# Supplementary Material for "End-to-end Learning of LDA by Mirror-Descent Back Propagation over a Deep Architecture"

## A   Derivation of $p(w_{d,1:N}|\theta_d, \Phi)$

To derive $p(w_{d,1:N}|\theta_d, \Phi)$, we first write $p(w_{d,1:N}, z_{d,1:N}|\theta_d, \Phi)$ as

$$p(w_{d,1:N}, z_{d,1:N}|\theta_d, \Phi) = \prod_{n=1}^{N} p(w_{d,n}|z_{d,n}, \Phi)p(z_{d,n}|\theta_d) \tag{16}$$

The expression $p(w_{d,1:N}|\theta_d, \Phi)$ can be evaluated in closed-form by marginalizing out $\{z_{d,n}\}_{n=1}^{N}$ in the above expression:

$$
\begin{aligned}
p(w_{d,1:N}|\theta_d, \Phi) &= \sum_{z_{d,1}} \cdots \sum_{z_{d,N}} \prod_{n=1}^{N} p(z_{d,n}|\theta_d) \cdot p(w_{d,n}|z_{d,n}, \Phi) \\
&= \prod_{n=1}^{N} \sum_{z_{d,n}} p(z_{d,n}|\theta_d) \cdot p(w_{d,n}|z_{d,n}, \Phi) \\
&= \prod_{n=1}^{N} \sum_{z_{d,n}} \left( \prod_{j=1}^{K} \theta_{d,j}^{z_{d,n,j}} \right) \left( \prod_{v=1}^{V} \prod_{j=1}^{K} \Phi_{vj}^{z_{d,n,j} \ w_{d,i,v}} \right) \\
&= \prod_{n=1}^{N} \sum_{z_{d,n}} \left( \prod_{v=1}^{V} \prod_{j=1}^{K} \theta_{d,j}^{z_{d,n,j}} \Phi_{vj}^{z_{d,n,j} \ w_{d,n,v}} \right) \\
&= \prod_{n=1}^{N} \left( \sum_{j=1}^{K} \theta_{d,j} \Phi_{vj} \right)^{w_{d,n,v}} \\
&= \prod_{v=1}^{V} \left( \sum_{j=1}^{K} \theta_{d,j} \Phi_{vj} \right)^{x_{d,v}}
\end{aligned}
\tag{17}
$$

where $w_{d,n,v}$ denotes the $v$-th element of the $V \times 1$ one-hot vector $w_{d,n}$, $w_{d,n}$ denotes the $n$-th word (token) inside the $d$-th document, and $x_{d,v}$ denotes the term frequency of the $v$-th word (in the vocabulary) inside the $d$-th document.

## B   Derivation of the Recursion for Mirror Descent Algorithm

First, we rewrite the optimization problem (11) as

$$\min_{\theta_d} \quad [\nabla_{\theta_d} f(\theta_{d,\ell-1})]^T (\theta_d - \theta_{d,\ell-1}) + \frac{1}{T_{d,\ell}} \Psi(\theta_d, \theta_{d,\ell-1}) \tag{18}$$

$$\text{s.t.} \quad \mathbb{1}^T \theta_d = 1, \quad \theta_d \succeq 0 \tag{19}$$

where $\theta_d \succeq 0$ denotes that each element of the vector $\theta_d$ is greater than or equal to zero. Using the fact that $\Psi(x, y) = x^T \ln(x/y) - \mathbb{1}^T x + \mathbb{1}^T y$, the constrained optimization problem (18)–(19) becomes

$$\min_{\theta_d} \quad [\nabla_{\theta_d} f(\theta_{d,\ell-1})]^T (\theta_d - \theta_{d,\ell-1}) + \frac{1}{T_{d,\ell}} \left[ \theta_d^T \ln \frac{\theta_d}{\theta_{d,\ell-1}} - \mathbb{1}^T \theta_d + \mathbb{1}^T \theta_{d,\ell-1} \right] \tag{20}$$

$$\text{s.t.} \quad \mathbb{1}^T \theta_d = 1, \quad \theta_d \succeq 0 \tag{21}$$

Dropping the terms independent of $\theta_d$, we can write (20)–(21) as

$$\min_{\theta_d} \quad [\nabla_{\theta_d} f(\theta_{d,\ell-1})]^T \theta_d + \frac{1}{T_{d,\ell}} \left[ \theta_d^T \ln \frac{\theta_d}{\theta_{d,\ell-1}} - \mathbb{1}^T \theta_d \right] \tag{22}$$

$$\text{s.t.} \quad \mathbb{1}^T \theta_d = 1, \quad \theta_d \succeq 0 \tag{23}$$

To solve (22)–(23), we write its Lagrangian as

$$L = [\nabla_{\theta_d} f(\theta_{d,\ell-1})]^T \theta_d + \frac{1}{T_{d,\ell}} \left[ \theta_d^T \ln \frac{\theta_d}{\theta_{d,\ell-1}} - \mathbb{1}^T \theta_d \right] + \lambda(\mathbb{1}^T \theta_d - 1) \tag{24}$$

where we relaxed the nonnegative constraint in the above Lagrange multiplier. However, we will show that the solution obtained will automatically be nonnegative mainly because of the logarithm term in the cost function. Taking the derivative of $L$ with respect to $\theta_d$ and $\lambda$ and setting them to zero, we have, respectively,

$$\frac{\partial L}{\partial \theta_d} = \nabla_{\theta_d} f(\theta_{d,\ell-1}) + \frac{1}{T_{d,\ell}} \left[ \ln \frac{\theta_d}{\theta_{d,\ell-1}} \right] + \lambda \mathbb{1} = 0$$

$$\frac{\partial L}{\partial \lambda} = \mathbb{1}^T \theta_d - 1 = 0$$

which leads to

$$\theta_d = \frac{\theta_{d,\ell-1} \odot \exp\left(-T_{d,\ell} \cdot \nabla_{\theta_d} f(\theta_{d,\ell-1})\right)}{\exp(T_{d,\ell} \cdot \lambda)}$$

$$\mathbb{1}^T \theta_d = 1$$

Solving the above two equations together, we obtain

$$\theta_d = \frac{1}{C_\theta} \theta_{d,\ell-1} \odot \exp\left(-T_{d,\ell} \cdot \nabla_{\theta_d} f(\theta_{d,\ell-1})\right) \tag{25}$$

where $C_\theta$ is a normalization factor such that $\theta_{d,\ell}$ adds up to one. Note that the above recursion can always guarantee non-negativity of the entries in the vector $\theta_{d,\ell}$ since we will always initialize the vector in the feasible region. Recall that $f(\theta_d)$ is the cost function on the right-hand side of (10), which is given by

$$f(\theta_d) = -x_d^T \ln(\Phi\theta_d) - (\alpha - \mathbb{1})^T \ln \theta_d$$

Therefore, the gradient of $f(\theta_d)$ can be computed as

$$\nabla_{\theta_d} f(\theta_d) = -\Phi^T \frac{x_d}{\Phi\theta_d} - \frac{\alpha - \mathbb{1}}{\theta_d} \tag{26}$$

Substituting the above gradient formula into (25), we obtain the desired result in (12).

## C   Implementation Details of the BP-sLDA

In this section, we describe the implementation details of the mirror-descent back propagation for the end-to-end learning of the supervised LDA model. Specifically, we will describe the details of the inference algorithm, and the model parameter estimation algorithm.

### C.1   Inference algorithm: Mirror Descent

Let $f(\theta_d)$ denote the objective function in (12). As we discussed in the paper, we use recursion (12) to iteratively find the MAP estimate of $\theta_d$ given $w_{d,1:N}$, which we repeat below:

$$\theta_{d,\ell} = \frac{1}{C_\theta} \cdot \theta_{d,\ell-1} \odot \exp\left(T_{d,\ell}\left[\Phi^T \frac{x_d}{\Phi\theta_{d,\ell-1}} + \frac{\alpha - \mathbb{1}}{\theta_{d,\ell-1}}\right]\right), \quad \ell = 1, \dots, L, \quad \theta_{d,0} = \frac{1}{K}\mathbb{1} \tag{28}$$

The step-size $T_{d,\ell}$ in mirror descent can be chosen to be either constant, i.e., $T_{d,\ell} = T$, or adaptive over iterations $\ell$ and documents $d$. To adaptively determine the step-size, we can use line search procedure. The inference algorithm with a simple line search can be implemented as Algorithm 1, where $\Psi(\theta_{d,\ell}, \theta_{d,\ell-1})$ can also be replaced by the squared vector 1-norm:

$$f(\theta_{d,\ell}) \leq f(\theta_{d,\ell-1}) + [\nabla_{\theta_d} f(\theta_{d,\ell-1})]^T (\theta_{d,\ell} - \theta_{d,\ell-1}) + \frac{1}{2T_{d,\ell}} \|\theta_{d,\ell} - \theta_{d,\ell-1}\|_1^2 \tag{29}$$

The line search approach determines the step-sizes adaptively, automatically stabilizing the algorithm and making inference converge faster. Moreover, the unsupervised model (BP-LDA) uses the same form of inference algorithm except that $\Phi$ is replaced with $\tilde{\Phi}$ and (27) is no longer needed.

**Algorithm 1** MAP Inference for BP-sLDA: Mirror-Descent with Line Search

1: Initialization: $\theta_{d,0} = \frac{1}{K}\mathbb{1}$ and $T_{d,0}$.
2: **for** $\ell = 1, \ldots, L$ **do**
3:     $T_{d,\ell} = T_{d,\ell-1}/\eta$, where $0 < \eta < 1$ (e.g., $\eta = 0.5$).
4:     **while** 1 **do**
5:        $\theta_{d,\ell} = \frac{1}{C_\theta} \cdot \theta_{d,\ell-1} \odot \exp\left(T_{d,\ell}\left[\Phi^T \frac{x_d}{\Phi\theta_{d,\ell-1}} + \frac{\alpha-\mathbb{1}}{\theta_{d,\ell-1}}\right]\right)$
6:        **if** $f(\theta_{d,\ell}) > f(\theta_{d,\ell-1}) + [\nabla_{\theta_d} f(\theta_{d,\ell-1})]^T (\theta_{d,\ell} - \theta_{d,\ell-1}) + \frac{1}{T_{d,\ell}}\Psi(\theta_{d,\ell}, \theta_{d,\ell-1})$ **then**
7:           $T_{d,\ell} \leftarrow \eta \cdot T_{d,\ell}$
8:        **else**
9:           break
10:        **end if**
11:     **end while**
12: **end for**
13: Inference result of $\theta_d$: $\theta_{d,L}$.
14: Inference result of $y_d$:

$$p(y_d|\theta_{d,L}, U, \gamma) = \begin{cases} N(U\theta_{d,L}, \gamma^{-1}) & \text{regression} \\ \text{Softmax}(\gamma U\theta_d) & \text{classification} \end{cases} \tag{27}$$

## C.2   Parameter Estimation: Stochastic Gradient Descent with Back Propagation

We first rewrite the training cost (14) as

$$J(U, \Phi) = \sum_{d=1}^{D} Q_d(U, \Phi) \tag{30}$$

where $Q_d(\cdot)$ denotes the loss function at the $d$-th document, defined as

$$Q_d(U, \Phi) \triangleq -\frac{1}{D} \ln p(\Phi|\beta) - \ln p(y_d|\theta_{d,L}, U, \gamma) \tag{31}$$

Note that, we do not have constraint on the model parameter $U$. Therefore, to update $U$, we can directly use the standard mini-batch stochastic gradient descent (SGD) algorithm. On the other hand, each column of the model parameter $\Phi$ is constrained to be on a $(V-1)$-dimension probability simplex, i.e, each element of $\Phi$ has to be nonnegative and each column sum up to one (i.e., $\Phi$ is a left-stochastic matrix). For this reason, we use stochastic mirror descent (SMD) to update each column of the model parameter $\Phi$, which is akin to the mirror descent algorithm for inference except that the gradient is replaced by stochastic gradient. The parameter estimation (learning) algorithm is described in Algorithm 2, where the expressions for the stochastic gradients $\frac{\partial Q_d}{\partial U}$ and $\frac{\partial Q_d}{\partial \Phi}$ are given in the next section. Note that we are allowing different columns of $\Phi$ to have different (and adaptive) learning rate, which makes the learning algorithm converge faster. This design is also akin to the construction in AdaGrad [8]. Finally, we also apply running average to the model parameters during SGD and SMD, which could improve the learning performance. In practical implementation, we could start the running average after after several passes of the training data.

## D   Gradient Formula of BP-sLDA

In this section, we give the gradient formula for the supervised learning of BP-sLDA. To this end, we first rewrite the training cost (14) as

$$J(U, \Phi) = \sum_{d=1}^{D} Q_d(U, \Phi) \tag{35}$$

where $Q_d(\cdot)$ denotes the loss function at the $d$-th document, defined as

$$Q_d(U, \Phi) \triangleq -\frac{1}{D} \ln p(\Phi|\beta) - \ln p(y_d|\theta_{d,L}, U, \gamma) \tag{36}$$

**Algorithm 2** Parameter Estimation for BP-sLDA: Stochastic Mirror Descent.

1: **for** $t = 1, 2, \ldots$ until converge **do**
2:     Sample a mini-batch of documents, denoted by $\mathcal{D}_t$.
3:     Infer $y_d$ and $\theta_d$ using Algorithm 1 for each document $d \in \mathcal{D}_t$.
4:     Compute the stochastic gradient $\partial Q_d / \partial U$ for $d \in \mathcal{D}_t$ according to (40).
5:     Compute the stochastic gradient $\partial Q_d / \partial \Phi$ for $d \in \mathcal{D}_t$ according to Algorithm 3.
6:     Compute the averaged stochastic gradient over $\mathcal{D}_t$:

$$\Delta U_t = \frac{1}{|\mathcal{D}_t|} \sum_{d \in \mathcal{D}_t} \frac{\partial Q_d}{\partial U}\Big|_{U=U_{t-1}, \Phi=\Phi_{t-1}}$$

$$\Delta \Phi_t = \frac{1}{|\mathcal{D}_t|} \sum_{d \in \mathcal{D}_t} \frac{\partial Q_d}{\partial \Phi}\Big|_{U=U_{t-1}, \Phi=\Phi_{t-1}}$$

    where $U_{t-1}$ and $\Phi_{t-1}$ denote the estimates of $U$ and $\Phi$ up to mini-batch $t-1$.
7:     Update $U$: $U_t = U_{t-1} - \mu_u \cdot \Delta U_t$.
8:     **for** each column $\phi_j$ of $\Phi$, $j = 1, \ldots, K$ **do**
9:         Set learning rate: $\mu_{\phi_j} = \mu_0 \Big/ \left( \sqrt{\frac{1}{t \cdot V} \sum_{\tau=1}^{t} \|\Delta \phi_{j,\tau}\|_2^2} + \epsilon \right)$
10:        Update $\phi_{j,t}$:

$$\phi_{j,t} = \frac{1}{C_{\phi_{j,t}}} \phi_{j,t-1} \odot \exp\left(-\mu_{\phi_j} \cdot \Delta\phi_{j,t}\right) \tag{32}$$

        where $C_{\phi_{j,t}}$ is a normalization factor that makes $\phi_{j,t}$ add up to one.
11:     **end for**
12:     Performing running average of the model parameters:

$$\bar{U}_t = \frac{t-1}{t} \bar{U}_{t-1} + \frac{1}{t} U_t \tag{33}$$

$$\bar{\Phi}_t = \frac{t-1}{t} \bar{\Phi}_{t-1} + \frac{1}{t} \Phi_t \tag{34}$$

13: **end for**
14: At convergence, $\bar{U}_t$ and $\bar{\Phi}_t$ will be final model parameters.

---

The expressions for the two terms in (36) are given by

$$-\frac{1}{D} \ln p(\Phi|\beta) = -\frac{1}{D} \ln \left( \left( \frac{\Gamma(V\beta)}{\Gamma(\beta)^V} \right)^K \prod_{j=1}^{K} \prod_{v=1}^{V} \Phi_{vj}^{\beta-1} \right)$$

$$= -\frac{1}{D} \sum_{j=1}^{K} \sum_{v=1}^{V} (\beta - 1) \ln \Phi_{vj} + \text{constant} \tag{37}$$

$$-\ln p(y_d | \theta_{d,L}, U, \gamma) = \begin{cases} -\sum_{j=1}^{V} y_{d,j} \ln \dfrac{\exp(\gamma \cdot p_{o,d,j})}{\sum_{m=1}^{C} \exp(\gamma \cdot p_{o,d,m})} & \text{classification} \\ \dfrac{1}{2\gamma} \|y_d - p_{o,d}\|_2^2 + \text{constant} & \text{regression} \end{cases}$$

$$= \begin{cases} -\sum_{j=1}^{C} y_{d,j} \gamma \cdot p_{o,d,j} + \ln \sum_{m=1}^{C} \exp(\gamma \cdot p_{o,d,m}) & \text{classification} \\ \dfrac{1}{2\gamma} \|y_d - p_{o,d}\|_2^2 + \text{constant} & \text{regression} \end{cases} \tag{38}$$

where $C$ in the above expressions is the number of output classes (in classification case), and

$$p_{o,d} \triangleq U\theta_{d,L} \tag{39}$$

**Algorithm 3** Mirror-Descent Back Propagation for BP-sLDA

1: Initialization of the error signal: $\xi_{d,L} = -(I - \mathbb{1}\theta_{d,L}^T) \cdot U^T \cdot \gamma(y_d - \hat{y}_d)$
2: **for** $\ell = L, \ldots, 1$ **do**
3: $\quad \xi_{d,\ell-1} = (I - \mathbb{1}\theta_{d,\ell-1}^T)\left\{ \frac{\theta_{d,\ell} \odot \xi_{d,\ell}}{\theta_{d,\ell-1}} - T_{d,\ell} \cdot \left[ \Phi^T \text{diag}\left( \frac{x_d}{(\Phi\theta_{d,\ell-1})^2} \right) \Phi + \text{diag}\left( \frac{\alpha-1}{\theta_{d,\ell-1}^2} \right) \right] (\theta_{d,\ell} \odot \xi_{d,\ell}) \right\}$
4: $\quad \Delta\Phi_{d,\ell} = T_{d,\ell} \cdot \left\{ \frac{x_d}{\Phi\theta_{d,\ell-1}} (\theta_{d,\ell} \odot \xi_{d,\ell})^T - \left[ \Phi(\theta_{d,\ell} \odot \xi_{d,\ell}) \odot \frac{x_d}{(\Phi\theta_{d,\ell-1})^2} \right] \theta_{d,\ell-1}^T \right\}$
5: **end for**
6: Compute the stochastic gradient $\partial Q_d / \partial\Phi$ according to:

$$\frac{\partial Q_d}{\partial\Phi} = -\frac{1}{D} \cdot \frac{\beta - 1}{\Phi} + \sum_{\ell=1}^{L} \Delta\Phi_{d,\ell} \tag{42}$$

Note that the choice of $p(y_d|\theta_{d,L}, U, \gamma)$ is not restricted to the above two options in our framework. Other forms could also be used and the corresponding gradient formula could also be derived. However, in sequel, we will only derive the gradient formula for these two classical choices.

### D.1 Gradient with respect to $U$

First, we derive the gradient of $Q_d(\cdot)$ with respect $U$. Note that the only term in (36) depending on $U$ is $\ln p(y_d|\theta_{d,L}, U, \gamma)$. Therefore, we have $\partial Q_d / \partial U = -\partial \ln p(y_d|\theta_{d,L}, U, \gamma)/\partial U$. Taking the gradient of (38) with respect to $U$ and after some simple algebra, we get

$$\frac{\partial Q_d}{\partial U} = \begin{cases} -\gamma \cdot (y_d - \hat{y}_d)\theta_{d,L}^T & \text{classification} \\ -\frac{1}{\gamma} \cdot (y_d - \hat{y}_d)\theta_{d,L}^T & \text{regression} \end{cases} \tag{40}$$

where $\hat{y}_d$ is defined as

$$\hat{y}_d = \begin{cases} \text{Softmax}(\gamma \cdot p_{o,d}), & \text{classification} \\ p_{o,d}, & \text{regression} \end{cases}$$
$$= \begin{cases} \text{Softmax}(\gamma \cdot U\theta_{d,L}), & \text{classification} \\ U\theta_{d,L}, & \text{regression} \end{cases} \tag{41}$$

### D.2 Gradient with respect to $\Phi$

In this subsection, we summarize the gradient expression for $\partial Q_d / \partial\Phi$ in Algorithm 3, where the derivation can be found in the next subsection. In Algorithm 3, $x_d$ and $y_d$ are the input bag-of-words vector and the label for the $d$-th document. The quantities $\theta_{d,\ell}$ and $\hat{y}_d$ are obtained and stored during the inference step, and the mirror-descent step-size $T_{d,\ell}$ is the one determined by line-search in the inference step (see Algorithm 1).

Similar to the inference in Algorithm 1, the above gradients can be computed efficiently by exploiting the sparsity of the vector $x_d$. For example, only the elements at the nonzero positions of $x_d$ need to be computed for $\Phi\theta_{d,\ell-1}$ and $\Phi(\theta_{d,\ell} \odot \xi_{d,\ell})$ since $\frac{x_d}{\Phi\theta_{d,\ell-1}}$ and $\frac{x_d}{(\Phi\theta_{d,\ell-1})^2}$ are known to be zero at these positions. Moreover, although $(\beta - 1)/\Phi$ is a dense matrix operation, it is the same within one mini-batch and can therefore be computed only once over each mini-batch, which can significantly reduce the amount of computation.

### D.3 Derivation of the gradient with respect to $\Phi$

In this subsection, we derive the gradient formula for $\Phi$. Note from (36) that, there are two terms that depend on $\Phi$, and

$$\frac{\partial Q_d}{\partial\Phi} = \frac{\partial}{\partial\Phi}\left( -\frac{1}{D} \ln p(\Phi|\beta) \right) + \frac{\partial}{\partial\Phi}\left( -\ln p(y_d|\theta_{d,L}, U, \gamma) \right) \tag{43}$$

The first term depends on $\Phi$ explicitly and its gradient can be evaluated as

$$\frac{\partial}{\partial \Phi}\left(-\frac{1}{D}\ln p(\Phi|\beta)\right) = \frac{\partial}{\partial \Phi}\left(-\frac{1}{D}\sum_{j=1}^{K}\sum_{v=1}^{V}(\beta-1)\ln \Phi_{vj}\right) = -\frac{1}{D}\cdot\frac{\beta-1}{\Phi} \tag{44}$$

The second term, however, depends on $\Phi$ implicitly through $\theta_{d,L}$. From Figure 2, we observe that $\theta_{d,L}$ not only depends on $\Phi$ explicitly (as indicated in the MDA block on the right-hand side of Figure 2) but also depends on $\Phi$ implicitly via $\theta_{d,L-1}$, which in turn depends on $\Phi$ both explicitly and implicitly (through $\theta_{d,L-2}$) and so on. That is, the dependency of the cost function on $\Phi$ is in a layered manner. For this reason, we need to apply chain rule to derive the its full gradient with respect to $\Phi$, which we describe below.

First, as we discussed above, each MDA block in Figure 2 contains $\Phi$, and $Q_d(U,\Phi)$ depends on the $\Phi$ appeared at different layers through $\theta_{d,L},\dots,\theta_{d,1}$. To derive the gradient formula, we first denote these $\Phi$ at different layers as $\Phi_L,\dots,\Phi_1$, and introduce an auxiliary function $R_d(U,\Phi_1,\dots,\Phi_L)$ to represent $-\ln p(y_d|\theta_{d,L},U,\gamma)$ with its $\Phi$ "untied" across layers in Figure 2. Then, the original $-\ln p(y_d|\theta_{d,L},U,\gamma)$ can be viewed as

$$-\ln p(y_d|\theta_{d,L},U,\gamma) = R_d(U,\Phi,\dots,\Phi) \tag{45}$$

where $\Phi_1 = \cdots = \Phi_L = \Phi$. Therefore, we have

$$\frac{\partial}{\partial \Phi}\left(-\ln p(y_d|\theta_{d,L},U,\gamma)\right) = \sum_{\ell=1}^{L}\frac{\partial R_d}{\partial \Phi_\ell}\Big|_{\Phi_\ell=\Phi} \tag{46}$$

where $\partial R_d/\partial \Phi_\ell$ denotes the gradient of $R_d(U,\Phi_1,\dots,\Phi_L)$ with respect to $\Phi_\ell$. Therefore, we only need to compute the gradient $\partial R_d/\partial \Phi_\ell$.

For simplicity of notation, we drop the subscript of $d$ in $\theta_{d,\ell}$. And since $\Phi$ is untied across layers in the mirror descent recursion (12) for the computation of $R_d(U,\Phi_1,\dots,\Phi_L)$, we can rewrite (12) as

$$z_\ell = T_{d,\ell}\cdot\left[\Phi_\ell^T\frac{x_d}{\Phi_\ell\theta_{\ell-1}} + \frac{\alpha-\mathbb{1}}{\theta_{\ell-1}}\right] \tag{47}$$

$$p_\ell = \theta_{\ell-1}\odot\exp(z_\ell) \tag{48}$$

$$\theta_\ell = \frac{p_\ell}{\mathbb{1}^T p_\ell} \tag{49}$$

where $z_\ell$ and $p_\ell$ are intermediate variables, and $\Phi$ is replaced with $\Phi_\ell$. To derive the gradient $\partial R_d/\partial \Phi_\ell$, it suffices to derive $\partial R_d/\partial \Phi_{\ell,ji}$. Note that

$$\frac{\partial R_d}{\partial \Phi_{\ell,ji}} = \frac{\partial p_\ell^T}{\partial \Phi_{\ell,ji}}\cdot\frac{\partial R_d}{\partial p_\ell} = \frac{\partial p_\ell^T}{\partial \Phi_{\ell,ji}}\cdot\delta_\ell \tag{50}$$

where

$$\delta_\ell \triangleq \frac{\partial R_d}{\partial p_\ell} \tag{51}$$

is an intermediate quantities that follows a backward recursion to be derived later. To proceed, we need to derive $\partial p_\ell^T/\partial \Phi_{\ell,ji}$:

$$\frac{\partial p_\ell^T}{\partial \Phi_{\ell,ji}} = \theta_{\ell-1}^T\odot\frac{\partial\exp(z_\ell^T)}{\partial \Phi_{\ell,ji}}$$

$$= \theta_{\ell-1}^T\odot\left[\frac{\partial z_\ell^T}{\partial \Phi_{\ell,ji}}\cdot\mathrm{diag}\big(\exp(z_\ell)\big)\right]$$

$$= \theta_{\ell-1}^T\odot\left[\frac{\partial z_\ell^T}{\partial \Phi_{\ell,ji}}\odot\mathbb{1}\exp(z_\ell^T)\right]$$

$$= \theta_{\ell-1}^T\odot\exp(z_\ell^T)\odot\frac{\partial z_\ell^T}{\partial \Phi_{\ell,ji}}$$

$$= p_\ell^T\odot\frac{\partial z_\ell^T}{\partial \Phi_{\ell,ji}} \tag{52}$$

Then, we need to derive the expression for $\partial z_l^T / \partial \Phi_{\ell,ji}$:

$$
\frac{\partial z_\ell^T}{\partial \Phi_{\ell,ji}} = T_{d,\ell} \cdot \left\{ \frac{\partial}{\partial \Phi_{\ell,ji}} \left( \frac{x_d^T}{\theta_{\ell-1}^T \Phi_\ell^T} \right) \cdot \Phi_\ell + \frac{x_d^T}{\theta_{\ell-1}^T \Phi_\ell^T} \cdot \frac{\partial \Phi_\ell}{\partial \Phi_{\ell,ji}} \right\}
$$

$$
= T_{d,\ell} \cdot \left\{ \frac{\partial}{\partial \Phi_{\ell,ji}} \left( \frac{x_d^T}{\theta_{\ell-1}^T \Phi_\ell^T} \right) \cdot \Phi_\ell + \frac{x_d^T}{\theta_{\ell-1}^T \Phi_\ell^T} \cdot E_{ji} \right\}
$$

$$
= T_{d,\ell} \cdot \left\{ -\frac{\partial \theta_{\ell-1}^T \Phi_\ell^T}{\partial \Phi_{\ell,ji}} \cdot \mathrm{diag}\left( \frac{x_d}{(\Phi_\ell \theta_{\ell-1})^2} \right) \cdot \Phi_\ell + \frac{x_d^T}{\theta_{\ell-1}^T \Phi_l^T} \cdot E_{ji} \right\}
$$

$$
= T_{d,\ell} \cdot \left\{ -\theta_{\ell-1}^T E_{ij} \cdot \mathrm{diag}\left( \frac{x_d}{(\Phi_\ell \theta_{\ell-1})^2} \right) \cdot \Phi_\ell + \frac{x_d^T}{\theta_{\ell-1}^T \Phi_\ell^T} \cdot E_{ji} \right\}
$$

$$
= T_{d,\ell} \cdot \left\{ -[\theta_{\ell-1}]_i \left[ \frac{x_d}{(\Phi_\ell \theta_{\ell-1})^2} \right]_j e_j^T \Phi_\ell + \left[ \frac{x_d}{\Phi_\ell \theta_{\ell-1}} \right]_j e_i^T \right\} \tag{53}
$$

where $e_i$ denotes the $i$-th natural basis vector in Euclidean space (i.e., the vector with the $i$-th element being one and all other element equal to zero), and $E_{ji}$ denotes a matrix whose $(j,i)$-th element is one and all other elements are zero. Substituting the above expression into (52), we obtain

$$
\frac{\partial p_\ell^T}{\partial \Phi_{\ell,ji}} = p_\ell^T \odot \frac{\partial z_\ell^T}{\partial \Phi_{\ell,ji}}
$$

$$
= T_{d,\ell} \cdot p_\ell^T \odot \left\{ -[\theta_{\ell-1}]_i \left[ \frac{x_d}{(\Phi_\ell \theta_{\ell-1})^2} \right]_j e_j^T \Phi_\ell + \left[ \frac{x_d}{\Phi_\ell \theta_{\ell-1}} \right]_j e_i^T \right\} \tag{54}
$$

Therefore,

$$
\frac{\partial R_d}{\partial \Phi_{\ell,ji}} = \frac{\partial p_\ell^T}{\partial \Phi_{\ell,ji}} \cdot \delta_\ell
$$

$$
= T_{d,\ell} \cdot p_\ell \odot \left\{ -[\theta_{\ell-1}]_i \left[ \frac{x_d}{(\Phi_\ell \theta_{\ell-1})^2} \right]_j e_j^T \Phi_\ell + \left[ \frac{x_d}{\Phi_\ell \theta_{\ell-1}} \right]_j e_i^T \right\} \delta_\ell
$$

$$
= T_{d,\ell} \cdot \left\{ -[\theta_{\ell-1}]_i \left[ \frac{x_d}{(\Phi_\ell \theta_{\ell-1})^2} \right]_j (p_\ell \odot e_j^T \Phi_\ell) \delta_\ell + \left[ \frac{x_d}{\Phi_\ell \theta_{\ell-1}} \right]_j (p_\ell \odot e_i^T) \delta_\ell \right\}
$$

$$
= T_{d,\ell} \cdot \left\{ -[\theta_{\ell-1}]_i \left[ \frac{x_d}{(\Phi_\ell \theta_{\ell-1})^2} \right]_j (p_\ell \odot e_j^T \Phi_\ell) \delta_\ell + \left[ \frac{x_d}{\Phi_\ell \theta_{\ell-1}} \right]_j [p_\ell]_i \cdot [\delta_\ell]_i \right\}
$$

$$
= T_{d,\ell} \cdot \left\{ -[\theta_{\ell-1}]_i \left[ \frac{x_d}{(\Phi_\ell \theta_{\ell-1})^2} \right]_j (e_j^T \Phi_\ell \mathrm{diag}(p_\ell)) \delta_\ell + \left[ \frac{x_d}{\Phi_\ell \theta_{\ell-1}} \right]_j [p_\ell]_i \cdot [\delta_\ell]_i \right\}
$$

$$
= T_{d,\ell} \cdot \left\{ -[\theta_l]_i \left[ \frac{x_d}{(\Phi_\ell \theta_{\ell-1})^2} \right]_j e_j^T \Phi_\ell (p_{l-1} \odot \delta_\ell) + \left[ \frac{x_d}{\Phi_\ell \theta_{\ell-1}} \right]_j [p_\ell]_i \cdot [\delta_\ell]_i \right\}
$$

$$
= T_{d,\ell} \cdot \left\{ -[\theta_{\ell-1}]_i \left[ \frac{x_d}{(\Phi_\ell \theta_{\ell-1})^2} \right]_j [\Phi_\ell (p_\ell \odot \delta_\ell)]_j + \left[ \frac{x_d}{\Phi_\ell \theta_{\ell-1}} \right]_j [p_\ell]_i \cdot [\delta_\ell]_i \right\} \tag{55}
$$

Writing the above expressions into matrix form (derivative with respect $\Phi_\ell$), we obtain:

$$
\frac{\partial R_d}{\partial \Phi_\ell} = T_{d,\ell} \cdot \left\{ \frac{x_d}{\Phi_\ell \theta_{\ell-1}} (p_\ell \odot \delta_\ell)^T - \left[ \Phi_\ell (p_\ell \odot \delta_\ell) \odot \frac{x_d}{(\Phi_\ell \theta_{\ell-1})^2} \right] \theta_{\ell-1}^T \right\} \tag{56}
$$

Now we need to derive the recursion for computing $\delta_\ell$. By the definition of $\delta_\ell$ in (51), we have

$$
\delta_{\ell-1} \triangleq \frac{\partial R_d}{\partial p_{\ell-1}}
$$

$$
= \frac{\partial \theta_{\ell-1}^T}{\partial p_{\ell-1}} \cdot \frac{\partial p_\ell^T}{\partial \theta_{\ell-1}} \cdot \frac{\partial R_d}{\partial p_\ell}
$$

$$
= \frac{\partial \theta_{\ell-1}^T}{\partial p_{\ell-1}} \cdot \frac{\partial p_\ell^T}{\partial \theta_{\ell-1}} \cdot \delta_\ell \tag{57}
$$

To continue, we have to evaluate $\frac{\partial \theta_{\ell-1}^T}{\partial p_{\ell-1}}$ and $\frac{\partial p_\ell^T}{\partial \theta_{\ell-1}}$. By (47)–(49), we have

$$
\frac{\partial p_\ell^T}{\partial \theta_{\ell-1}} = \frac{\partial \theta_{\ell-1}^T}{\partial \theta_{\ell-1}} \odot \mathbb{1} \exp(z_\ell^T) + \mathbb{1}\theta_{\ell-1}^T \odot \frac{\partial \exp(z_\ell^T)}{\partial \theta_{\ell-1}}
$$

$$
= I \odot [\mathbb{1}\exp(z_\ell^T)] + \mathbb{1}\theta_{\ell-1}^T \odot \left[ \frac{\partial z_\ell^T}{\partial \theta_{\ell-1}} \cdot \frac{\partial e_\ell^T}{\partial z_\ell} \right]
$$

$$
= \operatorname{diag}\big( \exp(z_\ell) \big) + \mathbb{1}\theta_{\ell-1}^T \odot \left[ \frac{\partial z_\ell^T}{\partial \theta_{\ell-1}} \cdot \operatorname{diag}\big( \exp(z_\ell) \big) \right]
$$

$$
= \operatorname{diag}\big( \exp(z_\ell) \big) + \mathbb{1}\theta_{\ell-1}^T \odot \left[ \frac{\partial z_\ell^T}{\partial \theta_{\ell-1}} \odot \mathbb{1}\exp(z_\ell^T) \right]
$$

$$
= \operatorname{diag}\big( \exp(z_\ell) \big) + \mathbb{1} \left[ \theta_{\ell-1}^T \odot \exp(z_\ell^T) \right] \odot \frac{\partial z_\ell^T}{\partial \theta_{\ell-1}}
$$

$$
= \operatorname{diag}\big( \exp(z_\ell) \big) + \mathbb{1}p_\ell^T \odot \frac{\partial z_\ell^T}{\partial \theta_{\ell-1}} \tag{58}
$$

To proceed, we need to derive the expression for $\frac{\partial z_\ell^T}{\partial \theta_{\ell-1}}$:

$$
\frac{\partial z_\ell^T}{\partial \theta_{\ell-1}} = T_{d,\ell} \cdot \left\{ \frac{\partial}{\partial \theta_{\ell-1}} \left( \frac{x_d^T}{\theta_{\ell-1}^T \Phi_\ell^T} \right) \Phi_\ell + \frac{\partial}{\partial \theta_{\ell-1}} \left( \frac{\alpha - \mathbb{1}}{\theta_{\ell-1}} \right)^T \right\}
$$

$$
= T_{d,\ell} \cdot \left\{ -\frac{\partial \theta_{\ell-1}^T \Phi_\ell^T}{\partial \theta_{\ell-1}} \cdot \operatorname{diag}\left( \frac{x_d}{(\Phi_\ell^T \theta_{\ell-1})^2} \right) \Phi_\ell - \operatorname{diag}\left( \frac{\alpha - \mathbb{1}}{\theta_{\ell-1}^2} \right) \right\}
$$

$$
= T_{d,\ell} \cdot \left\{ -\Phi_\ell^T \operatorname{diag}\left( \frac{x_d}{(\Phi_\ell^T \theta_{\ell-1})^2} \right) \Phi_\ell - \operatorname{diag}\left( \frac{\alpha - \mathbb{1}}{\theta_{\ell-1}^2} \right) \right\}
$$

$$
= -T_{d,\ell} \cdot \left\{ \Phi_\ell^T \operatorname{diag}\left( \frac{x_d}{(\Phi_\ell^T \theta_{\ell-1})^2} \right) \Phi_\ell + \operatorname{diag}\left( \frac{\alpha - \mathbb{1}}{\theta_{\ell-1}^2} \right) \right\} \tag{59}
$$

Substituting the above expression into (58), we get the expression for $\frac{\partial p_\ell^T}{\partial \theta_{\ell-1}}$:

$$
\frac{\partial p_\ell^T}{\partial \theta_{\ell-1}} = \operatorname{diag}\left\{ \exp\left( T_{d,\ell} \left[ \Phi_\ell^T \frac{x_d}{\Phi_\ell \theta_{\ell-1}} + \frac{\alpha - \mathbb{1}}{\theta_{\ell-1}} \right] \right) \right\}
$$

$$
- T_{d,\ell} \cdot (\mathbb{1}p_\ell^T) \odot \left[ \Phi_\ell^T \operatorname{diag}\left( \frac{x_d}{(\Phi_\ell \theta_{\ell-1})^2} \right) \Phi_\ell + \operatorname{diag}\left( \frac{\alpha - \mathbb{1}}{\theta_{\ell-1}^2} \right) \right]
$$

$$
= \operatorname{diag}\left( \frac{p_\ell}{\theta_{\ell-1}} \right) - T_{d,\ell} \cdot (\mathbb{1}p_\ell^T) \odot \left[ \Phi_\ell^T \operatorname{diag}\left( \frac{x_d}{(\Phi_\ell \theta_{\ell-1})^2} \right) \Phi_\ell + \operatorname{diag}\left( \frac{\alpha - \mathbb{1}}{\theta_{\ell-1}^2} \right) \right]
$$

$$
= \left\{ \operatorname{diag}\left( \frac{1}{\theta_{\ell-1}} \right) - T_{d,\ell} \cdot \left[ \Phi_\ell^T \operatorname{diag}\left( \frac{x_d}{(\Phi_\ell \theta_{\ell-1})^2} \right) \Phi_\ell + \operatorname{diag}\left( \frac{\alpha - \mathbb{1}}{\theta_{\ell-1}^2} \right) \right] \right\} \operatorname{diag}(p_\ell) \tag{60}
$$

To complete the derivation of the recursion (57), we need to derive $\frac{\partial \theta_{\ell-1}^T}{\partial p_{\ell-1,t}}$, which is given by

$$
\frac{\partial \theta_{\ell-1}^T}{\partial p_{\ell-1}} = \frac{\partial p_{\ell-1}^T}{\partial p_{\ell-1}} \cdot \frac{1}{\mathbb{1}^T p_{\ell-1}} + \frac{\partial}{\partial p_{\ell-1}} \left( \frac{1}{\mathbb{1}^T p_{\ell-1}} \right) p_{\ell-1}^T = \frac{I - \mathbb{1}\theta_{\ell-1}^T}{\mathbb{1}^T p_{\ell-1}} \tag{61}
$$

Expressions (57), (60) and (61) provide the complete backward recursion for $\delta_\ell$ from $\ell = L$ to $\ell = 1$. Finally, to initialize the backward recursion, we need the expression for $\delta_L$. By its definition,

we have

$$\delta_L \triangleq \frac{\partial R_d}{\partial p_L}$$

$$= \frac{\partial \theta_L^T}{\partial p_L} \cdot \frac{\partial p_{o,d}^T}{\partial \theta_L} \cdot \frac{\partial R_d}{\partial p_{o,d}}$$

$$= \frac{\partial \theta_L^T}{\partial p_L} \cdot U^T \cdot \frac{\partial R_d}{\partial p_{o,d}}$$

$$= \frac{1}{\mathbb{1}^T p_L}(I - \mathbb{1}\theta_L^T) \cdot U^T \cdot \frac{\partial R_d}{\partial p_{o,d}} \tag{62}$$

where in the last step we substituted (61). By (45) and (38), we have

$$\frac{\partial R_d}{\partial p_{o,d}} = \frac{\partial}{\partial p_{o,d}}\Big( -\ln p(y_d|\theta_{d,L}, U, \gamma)\Big)$$

$$= \begin{cases} -\gamma \cdot (y_d - \hat{y}_d) & \text{classification} \\ -\frac{1}{\gamma} \cdot (y_d - \hat{y}_d) & \text{regression} \end{cases} \tag{63}$$

Therefore,

$$\delta_L = \begin{cases} -\dfrac{1}{\mathbb{1}^T p_L}(I - \mathbb{1}\theta_L^T) \cdot U^T \cdot \gamma \cdot (y_d - \hat{y}_d) & \text{classification} \\ -\dfrac{1}{\mathbb{1}^T p_L}(I - \mathbb{1}\theta_L^T) \cdot U^T \cdot \dfrac{1}{\gamma} \cdot (y_d - \hat{y}_d) & \text{regression} \end{cases} \tag{64}$$

As a final remark, we found that in practical implementation $p_\ell$ could be very large while $\delta_\ell$ could be small, which leads to potential numerical instability. To address this issue, we introduce the following new variable:

$$\xi_{d,\ell} \triangleq \mathbb{1}^T p_\ell \cdot \delta_\ell \tag{65}$$

Then, the quantities $p_\ell$ and $\delta_\ell$ can be replaced with one variable $\xi_{d,\ell}$, and the backward recursion of $\delta_\ell$ can also be replaced with the backward recursion of $\xi_{d,\ell}$. Introducing $\Delta\Phi_\ell = \partial R_d/\partial \Phi_\ell$ and with some simple algebra, we obtain the back propagation and gradient expression for $\Phi$ in Algorithm 3.

## E  Gradient Formula of BP-LDA

The unsupervised learning problem (4) can be rewritten, equivalently, as minimizing the following cost function:

$$J(\tilde{\Phi}) = \sum_{d=1}^{D} Q_d(\tilde{\Phi}) \tag{66}$$

where $Q_d(\tilde{\Phi})$ is the loss function defined as

$$Q_d(\tilde{\Phi}) = -\frac{1}{D}\ln p(\tilde{\Phi}|\beta) - \ln p(w_{d,1:N}|\tilde{\Phi}, \alpha) \tag{67}$$

Taking the gradient of both sides of (67), we obtain

$$\frac{\partial Q_d}{\partial \tilde{\Phi}} = \frac{\partial}{\partial \tilde{\Phi}}\Big( -\frac{1}{D}\ln p(\tilde{\Phi}|\beta)\Big) + \frac{\partial}{\partial \tilde{\Phi}}\Big( -\ln p(w_{d,1:N}|\tilde{\Phi}, \alpha)\Big) \tag{68}$$

The first term in (68) has already been derived in (44):

$$\frac{\partial}{\partial \tilde{\Phi}}\ln p(\tilde{\Phi}|\beta) = \frac{\beta - 1}{\tilde{\Phi}} \tag{69}$$

where $\frac{\beta - 1}{\tilde{\Phi}}$ denotes elementwise division of the scalar $\beta - 1$ by the matrix $\tilde{\Phi}$. We now proceed to derive the second term in (68).

$$\frac{\partial}{\partial \tilde{\Phi}}\ln p(w_{d,1:N}|\tilde{\Phi}, \alpha) = \frac{1}{p(w_{d,1:N}|\tilde{\Phi}, \alpha)} \cdot \frac{\partial}{\partial \tilde{\Phi}}p(w_{d,1:N}|\tilde{\Phi}, \alpha)$$

$$= \frac{1}{p(w_{d,1:N}|\tilde{\Phi},\alpha)} \cdot \frac{\partial}{\partial\tilde{\Phi}} \int p(w_{d,1:N},\theta_d|\tilde{\Phi},\alpha)d\theta_d$$

$$= \frac{1}{p(w_{d,1:N}|\tilde{\Phi},\alpha)} \cdot \int \left[ \frac{\partial}{\partial\tilde{\Phi}} p(w_{d,1:N},\theta_d|\tilde{\Phi},\alpha) \right] d\theta_d$$

$$= \frac{1}{p(w_{d,1:N}|\tilde{\Phi},\alpha)} \cdot \int \left[ \frac{\partial}{\partial\tilde{\Phi}} \ln p(w_{d,1:N},\theta_d|\tilde{\Phi},\alpha) \right] \cdot p(w_{d,1:N},\theta_d|\tilde{\Phi},\alpha)d\theta_d$$

$$= \int \left[ \frac{\partial}{\partial\tilde{\Phi}} \ln p(w_{d,1:N},\theta_d|\tilde{\Phi},\alpha) \right] \cdot \frac{p(w_{d,1:N},\theta_d|\tilde{\Phi},\alpha)}{p(w_{d,1:N}|\tilde{\Phi},\alpha)}d\theta_d$$

$$= \int \left[ \frac{\partial}{\partial\tilde{\Phi}} \ln p(w_{d,1:N},\theta_d|\tilde{\Phi},\alpha) \right] \cdot p(\theta_d|w_{d,1:N},\tilde{\Phi},\alpha)d\theta_d$$

$$= \mathbb{E}_{\theta_d|w_{d,1:N}} \left[ \frac{\partial}{\partial\tilde{\Phi}} \ln p(w_{d,1:N},\theta_d|\tilde{\Phi},\alpha) \right] \tag{70}$$

Using (9), we rewrite $\ln p(w_{d,1:N},\theta_d|\tilde{\Phi},\alpha)$ as

$$\ln p(w_{d,1:N},\theta_d|\tilde{\Phi},\alpha) = \ln p(w_{d,1:N},\theta_d|\tilde{\Phi},\alpha)$$
$$= \ln p(w_{d,1:N}|\theta_d,\tilde{\Phi}) + \ln p(\theta_d|\alpha)$$
$$= \ln p(x_d|\theta_d,\tilde{\Phi}) + \ln p(\theta_d|\alpha) \tag{71}$$

Note that expression (70) applies expectation after taking the gradient with respect to $\tilde{\Phi}$. Therefore, the gradient of $\ln p(w_{d,1:N},\theta_d|\tilde{\Phi},\alpha)$ inside the expectation of (70) is taken by assuming that $\theta_d$ is independent of $\tilde{\Phi}$. Taking the gradient of both sides of (71) and using this fact, we obtain

$$\frac{\partial}{\partial\tilde{\Phi}} \ln p(w_{d,1:N},\theta_d|\tilde{\Phi},\alpha) = \frac{\partial}{\partial\tilde{\Phi}} \ln p(x_d|\theta_d,\tilde{\Phi}) \tag{72}$$

Substituting the above expression into (70), we obtain the desired result.