[Reviews · NeurIPS 2015]

Submitted by Assigned_Reviewer_1

=== Update after rebuttal ===

I thank the authors for a comprehensive rebuttal and extra experiments. It has addressed most of my concerns, and I have updated my score. The authors should make sure to properly tone down the claims about improved training for LDA (vs. sLDA), as well as mention the perplexity experiment. It seems to me that we do not really understand very well what is happening in these models at this stage; this perplexity experiment is just scratching the surface (and should be presented as such). I am also a bit puzzled by the use of alpha = 1.001 (vs. exactly 1 e.g.). As a prior, there is almost no difference between the two as a prior; but I can see that it changes a lot the MAP. In particular, using alpha = 1 (uniform distribution), the MAP is just maximum likelihood and could have zero entries; whereas any alpha > 1 creates a barrier in the objective function and prevents any theta_k to be too close to zero.

This being said, I think this paper makes interesting contributions, and so even though more careful experiments are needed to understand better what is happening, this can be left as future work and people can already build on the ideas introduced here for topic modeling.

== additional comments

[# of dominant topics experiment] The "# of dominant topics" experiment is interesting and should be included in the final version. As a note, a prior with alpha = 1.001 with K=100 gives on average about 60 dominant topics as a prior (defined as the number of topics to include to get 90% of the mass when sorted in decreasing order). This becomes 45 for alpha = 0.5 and 15 for alpha = 0.1 (which shows that these alpha are still too big to be used when K = 100). This experiment indicates that given the topic learned, the likelihood is still enforcing somewhat sparse topic proportion (I think that for an alpha so close to 1, the MAP should not be too different than the maximum likelihood solution). Though at K=100, one can already see a big difference in sparsity between BP-LDA alpha = 1.001 and Gibbs-LDA alpha=0.5.

[perplexity experiment] The authors should also report the perplexity for BP-LDA with alpha=0.5 and alpha=0.1 to control the difference in alpha. They should also try more topics. The reason that the perplexity is becoming worse with more topics is that alpha should change according to the number of topics. In my experience, properly estimating alpha yields a perplexity more robust to the number of topics (and can even get a better fit with more topics). To get some ball park, the alpha's learned by maximum likelihood using variational inference on standard text document datasets are much smaller than these values. For example, on the NIPS dataset from http://ai.stanford.edu/~gal/data.html, they are of the order of 1e-3 for a K=200 topic model. Finally, as I said in my previous review, it would interesting in future work to propagate the backprop to updating the alpha's as well...

=== end of update ===

This paper makes an interesting contribution to the supervised topic modeling literature. The idea of using backpropagation by unrolling a few steps of an iterative procedure (which computes a fixed point; or maximize some objective; or do some message passing) has been used before in several areas (as the cited [16]), but as far as I know, it has not been used for supervised topic modeling, so this is a fresh outlook, and I like it. It is also interesting that they are able to get significantly better results than linear regression (I hope this is ridge regression!), as usually supervised topic models struggle to improve over standard discriminative approaches. Their results in Figure 2b) where they basically do the same as logistic regression (within the variation) is more typical (though the other supervised topic model approaches do not do well on this one for some reasons).

* Missing experiment:

The reason that I am not putting a higher rating is because I think an important experiment is missing given that the authors claim that they their approximate MAP method to learn the parameters of the unsupervised LDA model is "outperforming previous learning methods". The classification / regression experiment is not meaningful for comparing unsupervised LDA learning techniques (it is great for sLDA, and the results are impressive; but unsupervised LDA is meant for an *unsupervised* evaluation!). The authors should thus also report the average test set per-word log-likelihood for LDA with the topic parameters learned by BP-LDA vs. Gibbs-LDA on their dataset. Note that they are definitively not allowed to do the PLSA-like technique of maximizing the posterior over theta on the test data to evaluate its likelihood (this is cheating!) -- they should just use the usual marginalization over theta for LDA. I suggest they use the code from http://homepages.inf.ed.ac.uk/imurray2/pub/09etm/ "Evaluation Methods for Topic Models", Wallach et al. ICML 2009, with some of their best methods. The authors should report these results in their rebuttal -- I am quite curious to see whether the superior prediction performance came at the cost of worse text modeling accuracy. If the authors can report on these results, I am willing to increase my rating (irrespective of whether their perplexity results are good or not).

* Novelty claim correction:

Using convex optimization to do MAP over theta for LDA is not new: this was already proposed in "Complexity of Inference in Latent Dirichlet Allocation", Sontag & Roy, NIPS 2011 (see Section 3.1). They also had proposed to use the exponentiated gradient algorithm to do the convex optimization, which is equivalent to mirror descent with KL divergence, i.e. yields exactly the same update as (12). The authors should properly correct this novelty claim and mention this prior work.

* Using alpha > 1 is not a good text model:

I also do not think that using alpha > 1 is a good idea for proper text modeling. It might be fine to get good classification / regression performance (it gives more dense features to the regressor); but if one would also evaluate the generative likelihood for the document, it won't be as good. All the work on topic modeling that I know always found that the individual hyperparameters of the Dirichlet prior over theta should be < 1 to give better perplexity (especially if a large number of topics is used). This was also mentioned in Section 3.2 of the [Sontag & Roy 2011] paper; see also Section 3 of

"On Smoothing and Inference for Topic Models", Asuncion et al. UAI 2009. I am willing to bet that if the authors do the perplexity experiment, they could get better results by using alpha < 1 (at least for the Gibbs sampler).

Also, note that it was also mentioned before that estimating the hyperparameters of the Dirichlet prior over theta (with asymmetric components) made a big difference for text modeling; see "Rethinking LDA: Why Priors Matter", Wallach et al., NIPS 2009. This was for unsupervised LDA. I am not sure whether it also could make a difference for sLDA -- but given that the authors have now an efficient backpropagation framework to learn all the parameters in a discriminative fashion; it might also be interesting to also backpropagate the gradient to estimate the alpha_k hyperparameters as well.

Using alpha > 1 means that a sampled theta is not sparse: this means that a specific document contains all topics with some probability, which certainly seems like a bad generative modeling assumption! I understand that the authors made this assumption because they wanted to claim that their MAP inference over theta was convex. On the other hand, given that they only do a small number L of iterations (and so they certainly do not optimize to convergence), and that the objective is not convex anyway in U and Phi, I do not think that having the inner MAP inference non-convex would be such a big problem. Using a finite number of iterations L is a way to lower bound the smallest value that theta could take in any case (as was suggested in Section 3.2 of [Sontag & Roy 2011]), and so the problem of negative infinities would operationally be avoided. They also seem to report better results for smaller alpha; so I would be curious to see Figure 4 to also report results for alpha < 1.

=== other comments ===

- Correction line 076: "this paper is the first work to perform a fully end-to-end discriminative training of LDA" -- it should be "of sLDA". They certainly do not do discriminative training of LDA: for LDA, they do an approximate MAP for Phi by removing the marginalization over theta that would normally be required by the model and replacing it with an approximate MAP over theta. This is still generative training, and so should not be called discriminative. In some sense, by doing MAP for theta, they are doing a kind of regularized PLSA model (probabilistic latent semantic analysis), which was the non-Bayesian precursor to LDA. In PLSA, there is a fixed theta per document, found by maximum likelihood. Here, they do regularized maximum likelihood for theta with the Dirichlet prior acting as a regularizer. It is presented as an approximation to LDA; but really, I would see it more as just regularized (potentially supervised) PLSA.

- Lines 120-122: the authors should clarify that Blei and McAuliffe had argued that using bar{z} instead of theta to tie the words to the variable to predict y was a better idea given their previous experience with two-signals modeling. I understand that the authors' framework could not handle bar{z} as the input variable for the distribution on y, as it is discrete, justifying their modification, but it is worthwhile to clarify this point. In their experiments, when they talk about "sLDA", do they mean the approach from [3] which used bar{z} as the input variable and also variational inference?

- Figure 3 a): are the error bars coming from different folds? What about the variation arising from just different random initialization for this non-convex problem? (Same thing for the sampling approach). Also, what about using a l2-regularizer on U to avoid overfitting for their method?

- Lines 396-399: do they use the same number of steps L when computing the test features; or they do full MAP for it?
Summary: [light reviewer]

quality:

6

(out of 10) clarity: 8 originality: 8 significance: 8

A very interesting paper. The exact MAP inference is not novel though (see below). And since they included their approximate MAP learning for unsupervised LDA as a contribution, they need to include a standard perplexity evaluation for it.

Submitted by Assigned_Reviewer_2

==update after rebuttal=== The authors do a good job performing new experiments which respond to most of my concerns and I have bumped up my score. The practical performance of \alpha < 1 is reassuring, and the effect of topic sparsity for \alpha = 1.001 is an interesting and new (to my knowledge) result, and should be presented in the final version with additional discussion. ==end update===

The authors derive a MAP objective for LDA, which for settings of Dirichlet hyperparameter \alpha > 1, is strictly concave and can be efficiently found, as opposed to the Bayesian posterior which is intractable. To handle the simplex constraints, the authors use a mirror descent algorithm with an entropic Bregman divergence which admits closed form updates. They formalize a fixed number of mirror descent iterations (later they show that the algorithm is not sensitive to this number) as a "layered architecture" and derive a backpropgation algorithm through this MDA algorithm. The backpropgation algorithm can then be applied to both unsupervised and supervised LDA objectives. The layered architecture representation is a useful abstraction makes it clear that any subsequent differentiable likelihood function can be used to connect the results of inference to observed data and obtain an efficient end-to-end algorithm. They evaluate their method against competing state of the art methods on two datasets, using both supervised and unsupervised variants and outperform the other methods in both predictive error and running time.

The paper is clearly written and easy to follow, including both derivations and experimental methodology. The introduction does a good job of situating the work among the rest of the literature. If anything, citations to related work are excessive and distracting, e.g. there are two MedLDA citations [20,21] with the same authors.

The AMR experiments are well done and easy to understand, particularly the runtime comparison

The main weaknesses I see in this paper which should be addressed in the rebuttal are, in priority order:

(1) In the MultiSent task, there is barely any improvement from Logistic regression. This means there is only one experiment (AMR) which convincingly shows that supervised LDA with the authors MAP-inference algorithm offers a significant boost in predictive performance. The authors should demonstrate the significance of their algorithm by obtaining a larger delta against bag-of-words regression on a larger number of datasets.

(2) The method requires a Dirichlet concentration of \alpha > 1, but this does not yield sparse distributions over topics for each documents, which is generally desirable in LDA. An experimental comparison sparsity for this method with \alpha > 1 and other methods for \alpha < 1 would be helpful, as well as an analysis of the impact of sparsity on predictive error. In addition, the experiments aggressively trimmed the vocabulary, which may mitigate the problem of non-sparse document-topic vectors. More experiments should be done where vocabulary size is varied.

(3) Compared to Bayesian methods, MAP may be prone to overfitting. The main regularization technique is to set the Dirichlet concentration parameters \alpha and \beta to be small positive values to encourage sparsity. Unfortunately this method limits choices here. How else can we mitigate overfitting in this model?

Summary: The authors propose a MAP inference method for LDA, relying on a convex objective function that can be solved with mirror descent. The inference method can be used to learn parameters for both supervised and unsupervised LDA and outperforms the existing state of the art in terms of both test error and runtime.

Submitted by Assigned_Reviewer_3

This paper proposes a novel inference procedure for doing MAP in supervised LDA. It seems like the main novelty of the paper is a procedure for learning the model in a (fully) discriminative fashion. The proposed approach yields empirical advantages compared to previous supervised LDA approaches on two tasks.

Overall I find this to be an interesting paper which proposes a novel way of training (supervised) topic models.

Some questions and comments: - Paper title: It would be worthwhile to emphasize that the paper is about

supervised LDA (even though "back prop" is a hint). I would also point

out that the approach proposes MAP inference. - In [3] the authors indicate that having the response variable be

dependent on the realized topic proportions (z) is advantageous. Is

there anything in your derivation that prevents you from doing the same?

It seems like you should be able to marginalize out theta. - Your approach requires the Dirichlet hyper-parameters to be greater than

1. Could you elaborate on the implication of this (e.g., on the sparsity

of the recovered topics and topic proportions)? It also appears that

competing methods such as sLDA are typically learned with alpha set to

values smaller than 1 (e.g., 1/K where K is the number of topics). This

should be compared to experimentally. - In the experiments I am not sure I see the advantage of reporting the

results on the smaller AMR dataset. I suggest only reporting results on

the full dataset (7.9M docs). Further, I think it's essential to report

the results of all baselines on the full dataset (it that's not possible

perhaps consider explaining why). - Overall the experiments seem to show that: 1) joint learning is

advantageous; 2) BP-LDA actually does better than vanilla inference

procedures for LDA. - Regarding 1) It would be interesting to further explain why joint

learning provides such an advantage. For example, are the learned topics

much different in joint methods vs. the others? My understanding is that

if one used BP-sLDA's topic proportions as covariates into linear

regression one could match BP-sLDA's results. Is this correct? My

intuition is that as the number of topics increase (it seems like using

100 topics for 7.9 million document may not be enough) BP-LDA and

BP-sLDA should do about the same. Have you noticed that in practice? - Regarding 2) Do you have an explanation as to why MAP (BP-LDA) does

better than fully-Bayesian inference (Gibbs-LDA)? Is it simply that it

finds a better local optimum? If so, as future work, it may be

interesting to explore how the ideas developed in this paper could be

used to optimize the variational objective. - Section 5.4 (computation time): It's useful to report computation time.

Comparing to other methods is also nice but the comparison seems unfair

as their code base is different.

Other comments: - In equation 11 what is \theta_d? - In the introduction readers may be led to believe that LDA is always

used for supervised task. I believe that is short-selling topic models.

It would be good to emphasize that the authors are interested in

versions of supervised topic models. - line 066: variable bound, did you mean 'variational'? - In your experiments did linear/logistic regression have access to all

words or only the top 5K? - The BP acronym is often used to denote belief propagation.

After rebuttal:

- Thanks for the rebuttal. It would be worthwhile to incorporate some of these points into the paper (specifically the extra dataset and the unsupervised results).

- In your rebuttal you wrote: "Joint learning enables the model to extract features most relevant to prediction, usually outperforming unsupervised learning + a separate."

classifier. "

I understand that. I think the size of the gap between joint training and unsupervised+classifier is still noteworthy. It would be good to provide an exploration of the topics.

- Thanks for your explanation of why you conditioned the response on the topics (and not z).
Summary: Good novel ideas, could have an impact. Experimental section is not completely convincing.

Submitted by Assigned_Reviewer_4

The paper proposed a new learning algorithm for supervised LDA, which is a hybrid of generative and discriminative methods.

The authors decoupled the parameter learning problem of supervised LDA into two optimization problems described in Eq. (3) and Eq. (4).

The proposed method iteratively compute $\Phi$ and $\theta_{d,L}$ and estimates $y_d$ using the learned $\theta_{d, L}$ by map inference.

The reviewer did not get the reason why the method computes $\hat{\phi}$ because $\hat{\phi}$ is never used to update $\theta_{d,L}$ and $\Phi$.
Summary: It is one of my light review papers. The paper proposed a new hybrid method for supervised LDA and empirically shows that it performs better score than existing methods.

Submitted by Assigned_Reviewer_5

Summary: The paper proposes a learning method for both unsupervised and supervised LDA, using exact MAP inference with mirror descent algorithm and backpropagation, instead of the traditional variational inference and Gibbs sampling approaches. Experimental results on predicting product review ratings (both regression and binary classification problems) show improvements over existing baselines.

The paper presents an interesting approach to learning traditional unsupervised/supervised LDA. I was particularly impressed with the scalability of the method, shown in Figure 5 where the training time of the proposed method on 7.9M documents is relatively comparable with that of the existing sLDA and MedLDA on 79K documents. However, besides that, I find that the arguments for using a fully discriminative training by MAP inference are not persuasive enough and are not supported well enough with empirical evidences. Here are my more detailed comments

One major concern I have with the proposed learning algorithm is that it only works for the case where \apha is greater than 1, as mentioned in section 3.1. I think this is a big limitation of this approach since in many settings, one desired property of LDA is its sparsity which is achieved by setting \sum_k \alpha_k < 1.

Second, in the experiments, the paper does not include details about the values of hyperparameters. Specifically, from my personal experience, \gamma plays a very important role in the prediction performance of sLDA. So, how do you choose \gamma in your experiments? Similar question applies to linear regression. I would also be good to include the performance of SVM.

In addition, I find Sections 3 and 4 describing the approach quite difficult to follow. Figure 2 is not particularly helpful. I would suggest providing a pseudo-code for the algorithm, in which some main steps can be explained in more details like in Section 3.1 and 3.2.

=== After rebuttal === I thank the authors for addressing my concerns and providing additional experimental results. I have ajusted my score accordingly.
Summary: The paper presents an interesting scalable learning method for both unsupervised and supervised LDA using exact MAP inference. However, I think that the proposed method has to trade-off an important property of traditional topic models (i.e., sparsity) while experimental results lack a lot of details to justify its effectiveness.

Author Feedback
Author rebuttal: We thank the reviewers for the valuable feedback. Below we address their concerns. Our revision will incorporate all points below plus other minor issues.

#alpha>1 vs alpha<1#
First, as pointed out by Reviewer 4, our method can also handle alpha<1 except that, in this case, the inference is no longer convex and hence no global optimal MAP inference is guaranteed as for the methods prior to this work. We did not pay much attention to alpha<1 earlier because we wanted to focus on convex inference. Based on all reviewers' feedback, we did new experiments and provide the resulting pR^2 on AMR (7.9M) below. They are not as good as our reported scores with alpha>1, but still significantly beats baselines.
#topics: 5 10 20 50 100
BP-sLDA (alpha=0.5): 0.488 0.548 0.575 0.571 0.574
BP-sLDA (alpha=0.1): 0.441 0.558 0.572 0.569 0.570

Second, our main contribution is to develop mirror-descent BackProp to improve prediction performance of supervised LDA. For fair evaluation, we first built strong baselines whose performance is tuned carefully (& indeed this led to alpha<1 for conventional topic models). Then the experimental results show that our method with alpha>1 still significantly outperforms all baseline topic models. To corroborate further, we carried out new large-scale experiments with BP-sLDA on AMR (7.9M) with #topics=200&alpha=1.001. We obtain pR^2=0.74, which even significantly beats a 200-unit neural network (pR^2=0.64) with the same model size.

Finally, when alpha is close to 1 (e.g. 1.001), the topic distribution can still be made reasonably sparse, as the probability simplex constraint on theta is akin to L1 regularization. To quantitatively demonstrate this, we carried out new experiments which gave the average # dominant topics (which add up to 90% probability) on AMR:
#topics: 5 10 20 50 100
BP-sLDA (alpha=1.001): 2.1 3.0 3.8 4.5 6.9
BP-LDA (alpha=1.001): 1.5 2.4 3.6 6.0 8.7
Gibbs-LDA (alpha=0.5): 1.6 1.7 1.9 2.1 2.3
Gibbs-LDA (alpha=0.1): 1.5 1.7 1.8 2.1 2.2

To Reviewer 1:
In Fig.3b (MultiSent), AUC=90.4% for LR, and AUC=91.3%, 91.2%, 91.4%, 90.8% for BP-sLDA with #topics=5, 10, 20, 50, 100. Note 1% absolute AUC improvement over 90.4% is significant (10% relative improvement). In revision, we will produce better visualization with error bars more clearly shown. In any case, BP-sLDA significantly beats prior-art topic models (AUC<80%).

Following your suggestion, we conducted a new binary text classification experiment with highly promising results on a large-scale proprietary dataset for business-centric applications (1.2M documents & 128K vocab), to be released in full after successful patent filing. In this new task, BP-sLDA (200 topics) achieves AUC=92.2% & error rate=15.2%, while LR has AUC=90.5% & error rate=17.1% (11% relative error rate cut). The gain is consistent with what was reported in the original draft, addressing your concern.

We set AMR's vocabulary size to 5k to be consistent with [18]. The pR^2 for full 701K-vocab AMR (7.9M) are:
#topics: 5 10 20 50 100
BP-sLDA (alpha=1.001): 0.633 0.677 0.672 0.682 0.684
Linear Regression: 0.403
even better than our reported results (5K-vocab AMR, 7.9M) in Table 1, showing that our method can effectively handle overfitting.

To Reviewer 2:
We make y depend on theta instead of z because it leads to a differentiable cost trainable by BackProp & SGD (z: discrete).

Joint learning enables the model to extract features most relevant to prediction, usually outperforming unsupervised learning + a separate classifier.

We conjecture that high performance of our method comes from better local optimum. Theoretical justification is our future work.

Running other LDA baselines on AMR (7.9M) takes too long (estimated to be ~10,000hrs shown in Fig.5).

To Reviewer 3:
We'll release codes with config files including all hyper-parameters.

In our model, gamma=1. After tuning, we found that performance is insensitive to gamma when gamma>=1.

The revised version will add the pseudo-code.

SVM: AUC=89.3% on MultiSent. SVR: pR^2=0.385 on AMR(7.9M).

To Reviewer 4:
Thanks for the reference [S&R] on convex inference & mirror-descent, which we will cite properly in the revised paper. To clarify, our contribution is to integrate BackProp with MDA to improve prediction performance, while [S&R] studies inference alone.

sLDA is from [3].

L is the same for training & test.

We appreciate your willingness to increase the rating. Indeed, our new experiment on per-word log-likelihoods (using [Wallach]'s code) confirms your bet:
#topics: 5 10 20
BP-LDA (alpha=1.001): -6.83 -7.12 -7.41
Gibbs (alpha=0.5): -6.17 -6.18 -6.21
Gibbs (alpha=0.1): -6.15 -6.13 -6.21

To Reviewer 5:
By \hat{\phi}, I assume you meant \tilde{\Phi}. We decouple learning into supervised & unsupervised parts, which learn \Phi & \tilde{\Phi} separately.

To Reviewer 6:
See our responses in section "alpha>1 vs alpha<1".